# Random quench predicts universal properties of amorphous solids

Masanari Shimada

*Graduate School of Arts and Sciences, The University of Tokyo, Tokyo 153-8902, Japan*

Eric De Giuli

*Department of Physics, Ryerson University, M5B 2K3, Toronto, Canada*

Amorphous solids display numerous universal features in their mechanics, structure, and response. Current models assume heterogeneity in mesoscale elastic properties, but require fine-tuning in order to quantitatively explain vibrational properties. A complete model should derive the magnitude and character of elastic heterogeneity from an initially structureless medium, through a model of the quenching process during which the temperature is rapidly lowered and the solid is formed. Here we propose a field-theoretic model of a quench, and compute structural, mechanical, and vibrational observables in arbitrary dimension $d$. This allows us to relate the properties of the amorphous solid to those of the initial medium, and to those of the quench. We show that previous mean-field results are subsumed by our analysis and unify spatial fluctuations of elastic moduli, long-range correlations of inherent state stress, universal vibrational anomalies, and localized modes into one picture.

## I. INTRODUCTION

At Kelvin-scale temperatures, glasses universally present mechanical and vibrational anomalies with respect to crystalline solids: below $1K$, the heat capacity behaves as $C(T) \propto T$, to be compared with $C(T) \propto T^3$ for crystals [1]. It is accepted that the anomalous behavior of $C(T)$ in glasses is caused by quantum mechanical tunneling between nearby energy minima in phase space, the two-level systems (TLS) initially proposed as a phenomenological model [2]. Recently, a microscopic picture of the TLS has begun to emerge, thanks to numerical simulations in which quasi-localized vibrational modes have been identified and counted [3–9].

Moreover, near $10K$, glasses display an excess of vibrational modes over phonons, the so-called 'boson peak.' It is not agreed what is the cause of the modes near the boson peak: the glass-specific behavior has variously been attributed to soft localized modes [10, 11], generic stiffness disorder [12, 13], proximity to jamming [14–16], and proximity to elastic instability [16].

The jamming approach predicts a regime in which the density of vibrational states $g(\omega)$ scales as $g(\omega) \propto \omega^2$ in any dimension $d$, sufficiently close to jamming *and* elastic instability [16]. The corresponding modes are extended but not plane waves, instead showing vortex-like motion at the particle scale. This law has been confirmed in numerical simulations [16–19], but is found to break down below some frequency $\omega_0$, below which there are only phonons and quasi-localized modes [7, 17, 19]. The latter, now confirmed to be present in many glass models, has a density following $g(\omega) \propto \omega^\alpha$ where $3 \leq \alpha \leq 4$ [3–8, 19–22]. Some authors argue that $\alpha = 4$ in the thermodynamic limit [22], while others argue that smaller exponents are possible due to interactions between localized instabilities [8]. Microscopic theory is needed to clarify these results.

The frequency $\omega_0$ setting the lower-limit of the jamming regime is controlled by the distance to elastic instability [16]. It was proposed that glasses dominated by repulsive interactions lay close to elastic instability due to the quench dynamics [15, 16]. This suggests that a model faithful to the physics of the quench might shed light on the density of quasi-localized modes and the frequency $\omega_0$ below which they become important. Ideally, any such model should also reproduce the universal vibrational features captured in prevous models [13, 16], such as the dip in sound speed and crossover in acoustic attenuation observed in many experiments [23–26].

Here we present such a model. Following a crude but principled model of an overdamped quench into an inherent state (IS), we derive universal vibrational properties characterized by complex elastic moduli and the density of vibrational states. We recover the exact equations governing mean-field vibrational properties discussed previously [16]. As a bonus, our model predicts other universal mechanical features, namely short-range correlations in elastic moduli and long-range correlations in the IS stress, as observed recently in simulations [27–32] and experiments [33][34]. Ultimately, the model fails to predict the universal $\omega^4$ law discussed above, thus indicating the minimal features needed to recover mean-field results, while highlighting routes towards a more complete model.

## II. RANDOM QUENCH MODEL

Consider the elasticity equation

$$\rho \frac{d^2 \vec{u}_i}{dt^2} - \partial_j \left[ S_{ijkl} \partial_k \vec{u}_l \right] = \vec{F}_i, \tag{1}$$

where $\rho$ is density, $\vec{u}_i$ is a displacement field, and $S_{ijkl} = C_{ijkl} + \delta_{ik}\sigma_{jl}$ in terms of the elastic modulus tensor $C_{ijkl}$ and the IS stress $\sigma_{jl}$. The elastic Green's function $G_{ij}$ is the solution to (1) for a Dirac-delta function forcing, $\vec{F}_j(\vec{r}) = \vec{f}_j\delta(\vec{r})$, that is, $\vec{u}_i(\vec{r}) = G_{ij}(\vec{r}) \cdot \vec{f}_j$. This is one of the fundamental observables in linear elasticity and captures many properties of linear response, including sound speed, damping due to the disorder, and the density of vibrational states, as discussed below.

At the mesoscale, the elastic moduli and the stress tensor $\sigma_{ij}$ can be considered to be spatially fluctuating fields. Vibrational properties can be derived from the disorder-averaged Green's function $\overline{G}_{ij}$, for different models of random disorder. The model of Schirmacher corresponds to random Gaussian fluctuations of elastic moduli [12, 13].

From the jamming approach, Ref. [16] employed a microscopic lattice model, which does not directly correspond to (1). However, elastic instability is caused by the destabilizing effect of stress in particulate matter with short-range repulsive interactions [15, 16]. Its proposed importance highlights stress as an important control parameter for vibrational properties. Moreover, it has been shown that the stress also plays a crucial role in the emergence of quasi-localized modes [5]. These works suggest that one should consider a model in which the IS stress $\sigma_{ij}$ is randomly fluctuating. Such an effort must immediately confront a no-go theorem of Di Donna and Lubensky [35]. In a comprehensive treatment of non-affine correlations in random media, the latter authors showed that a random IS stress alone does *not* yield non-affine motion, and therefore cannot give rise to anomalous vibrational properties: a material that behaves affinely is a homogeneous continuum, the continuum limit of a crystal. How can we reconcile the importance of destabilizing stress with its apparently mild effect on non-affine motion?

We propose that the solution is to consider the quench itself. Indeed, as shown in [35], if random forces are added to an initially featureless continuum, then the relaxation to an IS will produce both a random IS stress and fluctuations in elastic moduli. The latter cause anomalous vibrational properties.

To probe how fluctuations in stress and elastic moduli are created during the final descent into an IS, we consider the following idealized model of a quench. We begin with the dense liquid in its natural state at the parent temperature $T_0$. This state is characterized by a stress tensor field $\tilde{\sigma}_{ij}(\vec{r})$, which we call the quench stress. In the case of a pair potential, this corresponds to (e.g. [36])

$$\tilde{\sigma}_{ij}(\vec{r}) = \sum_a \delta(\vec{r} - \vec{r}_a) \left[ mv_i^a v_j^a + \tfrac{1}{2} \sum_{b \neq a} r_i^{ab} f_j^{ab} \right] \tag{2}$$

where $\vec{v}^a$ is the velocity of particle $a$, $\vec{r}^{ab} = \vec{r}^a - \vec{r}^b$, and $\vec{f}^{ab}$ is the force exerted by particle $b$ on particle $a$.

From this state we then instantaneously quench the temperature to 0, so that the velocity field vanishes. Since the state is not in mechanical equilibrium, it will relax under the action of the quench stress $\tilde{\sigma}_{ij}$ until it finds a state of mechanical equilibrium, as depicted in Figure 1. The process ends as soon as a state of mechanical equilibrium is found. We will then compute the disorder-averaged Green's function of the IS.

Since our aim is to model the emergence of structure in the glass, we wish to consider the simplest possible model of the dense liquid. Only the short-time response of the liquid is relevant, so we consider the latter as an initial homogeneous elastic continuum with elastic constants $\tilde{\lambda}$ and $\tilde{\mu}$, which gives the elastic modulus tensor $C_{ijkl} = \tilde{\lambda}\delta_{ij}\delta_{kl} + \tilde{\mu}(\delta_{ik}\delta_{jl} + \delta_{il}\delta_{jk})$. We assume, for simplicity, that these constants are spatially uniform. When the temperature is removed, the continuum deforms under the quench stress, just as a crumpled elastic membrane will relax when external constraints are removed. For simplicity we ignore inertia during the quench, so our model is one of an overdamped quench. This is equivalent to gradient descent as performed in numerical simulations.

Although crude, this model is perhaps the simplest that captures the final stages of descent into an IS; more advanced models could allow the liquid to have non-trivial structural correlations and consider inertia in the quenching process.

We aim to describe universal properties of this process. Since we work in the continuum, the relevant distribution of $\tilde{\sigma}_{ij}(\vec{r})$ can be inferred using field-theoretical arguments [37, 38]. Indeed, under our assumption of a featureless liquid, under coarse-graining the quench stress field will retain only short-range correlations. As we eventually seek the small $q$ behaviour of the solid, we take this small length scale to 0 immediately to obtain a simple Gaussian distribution of the symmetric tensor field $\tilde{\sigma}_{ij}(\vec{r})$:

$$\mathbb{P}[\tilde{\sigma}] \propto \exp\left( -\tfrac{1}{2} \int_r \left[ s_1 \tilde{\phi}_{ij} \tilde{\phi}_{ij} + s_2 \tilde{\sigma}_{ii} \tilde{\sigma}_{jj} \right] \right), \tag{3}$$

where $\tilde{\phi}_{ij} = \tilde{\sigma}_{ij} - \tfrac{1}{d}\delta_{ij}\tilde{\sigma}_{kk}$ is the deviatoric stress, and $s_1$ and $s_2$ are parameters related to the magnitude of quench stress by

$$\overline{\tilde{\phi}_{ij}(\vec{r})\tilde{\phi}_{ij}(0)} = \frac{d^2 - 1}{s_1}\delta(\vec{r}), \tag{4}$$

$$\overline{\tilde{\sigma}_{ii}(\vec{r})\tilde{\sigma}_{jj}(0)} = \frac{1}{s_2}\delta(\vec{r}) \tag{5}$$

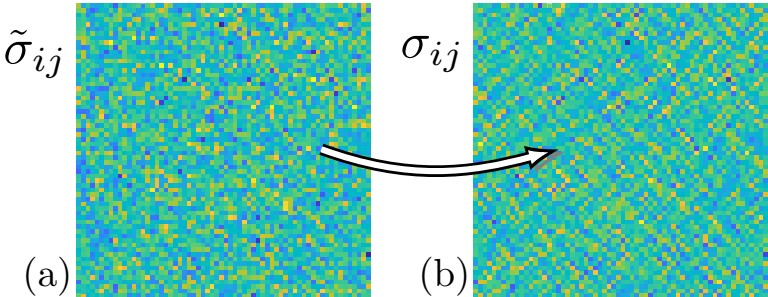

FIG. 1. Illustration of random quench model. From (a) an initial disordered state with quench stress $\tilde{\sigma}_{ij}$, we allow the system to relax to an inherent state with stress $\sigma_{ij}$, (b). Illustrated here are $\tilde{\sigma}_{xy}$ and $\sigma_{xy}$ for $\mu/\lambda = 0.1$. The long-range correlations in $\sigma_{xy}$ are apparent.

in $d$ dimensions. The constant component of the quench stress will be treated separately.

Standard renormalization arguments [37, 38] predict that corrections to (3) will introduce a length scale $a$ which is on the order of the relevant microscopic length, namely the particle radius. We therefore expect our model to be valid on scales much larger than this size. In particular, the Gaussian distribution (3) should be valid, so long as the liquid state does not have relevant long-range correlations. In our continuum treatment, we will employ a cutoff $\Lambda \approx 2\pi/a$ in momentum space, so that corrections to (3) need not be explicitly incorporated.

Consider the displacement field $u_i(\vec{r})$ along the quench. At any moment, there is an instantaneous stress field $\sigma_{ij}[\vec{u}](\vec{r})$. In an overdamped quench, a new IS will be found as soon as $\sigma_{ij}$ is in mechanical equilibrium. (If inertia were considered, the system could jump over shallow minima before finding a resting state.) Di Donna and Lubensky found the new IS stress $\sigma_{ij}(\vec{r})$ and the elastic modulus tensor $C'_{ijkl}(\vec{r}) = C_{ijkl} + \delta C_{ijkl}(\vec{r})$ around the IS, to leading order in $u_i$, for arbitrary quench stress fields $\tilde{\sigma}_{ij}(\vec{r})$ with zero spatial average. The result is a pair of linear functionals

$$\delta C_{ijkl}(\vec{q}) = \mathscr{S}_{ijklmn}(\vec{q})\tilde{\sigma}_{mn}(\vec{q}) \tag{6}$$

$$\sigma_{ij}(\vec{q}) = \mathscr{P}_{ijkl}(\vec{q})\tilde{\sigma}_{kl}(\vec{q}) \tag{7}$$

in Fourier space, where $\mathscr{S}$ and $\mathscr{P}$ depend on the elastic moduli and the momentum $\vec{q}$. Explicitly, these take the form

$$\mathscr{S}_{ijklmn}(\vec{q}) = -\frac{1}{2(1-c)}[2c(\delta_{ij}\delta_{ke}\delta_{lf} + \delta_{kl}\delta_{ie}\delta_{jf}) + (1-c)(\delta_{ik}\delta_{je}\delta_{lf} + \delta_{jl}\delta_{ie}\delta_{kf} + \delta_{il}\delta_{je}\delta_{kf} + \delta_{jk}\delta_{ie}\delta_{lf})]V_{efmn} \tag{8}$$

$$\mathscr{P}_{ijkl}(\vec{q}) = \frac{1}{2}P^T_{ik}P^T_{jl} + \frac{1}{2}P^T_{il}P^T_{jk} - \frac{1}{1+2\mu}P^T_{ij}\hat{q}_k\hat{q}_l, \tag{9}$$

where $c = 1/(1+2\mu)$, $P^T_{ij} = \delta_{ij} - \hat{q}_i\hat{q}_j$, and

$$V_{efmn} = \hat{q}_e(\delta_{fm}\hat{q}_n + \delta_{fn}\hat{q}_m) + \hat{q}_f(\delta_{em}\hat{q}_n + \delta_{en}\hat{q}_m) - 2(1+c)\hat{q}_e\hat{q}_f\hat{q}_m\hat{q}_n. \tag{10}$$

Ref. [35] did not include any constant component of the quench stress, but this is easily added. To leading order in $u$, we find that if $\widetilde{\sigma}_{ij}(\vec{r}) = \bar{p}\delta_{ij}$, then this is equivalent to replacing the Lamé moduli by $\lambda = \tilde{\lambda} - d\bar{p}$ and $\mu = \tilde{\mu} + \bar{p}$.

### A. Stress correlations & elastic moduli fluctuations:

Combining Eqs.(6),(7) with (3) immediately yields predictions for the distribution of local elastic moduli and the distribution of IS stress.

The derivation of the IS stress distribution is nontrivial because of mechanical equilibrium, which is satisfied by $\sigma$ but not by $\tilde{\sigma}$. The derivation is sketched in Appendix 1.

The result is conveniently represented in terms of a gauge field [37–39]. In $d = 2$ we can write $\sigma_{ij} = \epsilon_{ik}\epsilon_{jl}\partial_k\partial_l\psi$, where $\epsilon$ is the antisymmetric tensor with $\epsilon_{12} = -\epsilon_{21} = 1$ and $\epsilon_{11} = \epsilon_{22} = 0$, and we predict

$$\mathbb{P}[\sigma[\psi]] \propto \exp\left(-\tfrac{1}{2}\tilde{\eta}\int_r \text{tr}^2\sigma\right), \quad d = 2, \tag{11}$$

with

$$\tilde{\eta} = \frac{(1+2\mu)^2 s_1 s_2}{4(1+\mu)^2 s_2 + 2\mu^2 s_1}, \quad d = 2. \tag{12}$$

Similarly, in $d = 3$ we can write $\sigma_{ij} = \epsilon_{ikl}\epsilon_{jmn}\partial_k\partial_m\Psi_{ln}$ and we predict

$$\mathbb{P}[\sigma[\Psi]] \propto \exp\left(-\frac{1}{2}\int_r \left[\eta\, \mathrm{tr}^2(\sigma) + g\, \mathrm{tr}\sigma^2\right]\right), \quad d = 3, \tag{13}$$

where

$$\eta = -\frac{s_1}{2}\frac{4\mu^2 s_1 + (9 - 12\mu^2)s_2}{8\mu^2 s_1 + 3(3 + 2\mu)^2 s_2} \quad d = 3 \tag{14}$$

$$g = \frac{s_1}{2}. \tag{15}$$

Eq.(11) and Eq.(13) are in precise agreement with [37, 38] when boundary effects are neglected. We emphasize that $\sigma$ is a functional of $\psi$ in $d = 2$ and $\Psi$ in $d = 3$ and thus these distributions are nontrivial. As shown in Appendix 1, these gauge fields are necessary to enforce the constraints on the stress tensor $\sigma^T = \sigma$ and $\nabla \cdot \sigma = 0$. They predict anisotropic long-range correlations in the stress field, as discussed at length in [37, 38].

We can also determine the distribution of local elastic moduli. We focus here on the bulk modulus fluctuation $\delta K = \delta C_{iikk}/d^2$ and shear modulus fluctuation $\delta\mu = [d\delta C_{ijij} - \delta C_{iijj}]/(d^3 + d^2 - 2d)$. These are predicted to be Gaussian, with fluctuations

$$\langle\delta K(\vec{r})\delta K(0)\rangle = \frac{2}{s_1 d^4}\left[\bar{d} + (5c\bar{d} + 4)^2 + \frac{s_1 - ds_2}{d^2 s_2}(d + 1 + 5c\bar{d})^2\right]\delta(\vec{r}) \tag{16}$$

$$\langle\delta\mu(\vec{r})\delta\mu(0)\rangle = \frac{2}{s_1 d^2(d+2)^2\bar{d}}\left[(d^2 + 1)^2 + \bar{d}(2d + 4 + c(d^2 - 2d - 5))^2\right. \tag{17}$$

$$\left. +\bar{d}(d^2 - 2d + 5 + c(d^2 - 2d - 5))^2\frac{s_1 - ds_2}{d^2 s_2}\right]\delta(\vec{r})$$

with $\bar{d} = d - 1$ and $c = 1/(1 + 2\mu)$. The strictly local nature of these correlations is a consequence of the local quench stress correlations. We expect corrections to these correlations only at the particle scale, as observed in generic models [27, 29].

Eqs. (12), (14), (15), (16), and (17) can be used to relate the strength of stress correlations and elastic moduli fluctuations to the quench stress, and to each other.

### B. Green's function & effective medium theory:

We now proceed to determine the Green's function $\overline{G}_{ij}$. We consider response at frequency $\omega$ and write the elasto-dynamic equation as a tensorial linear operator $\hat{A}(\omega; \sigma)$ acting on $\vec{u}$,

$$A_{il}(\omega) \equiv -\rho\omega^2\delta_{il} - (\partial_j S_{ijkl})\partial_k - S_{ijkl}\partial_j\partial_k$$

We need to compute the Green's function $\hat{G}(\vec{r}; \vec{r}_0)$, the solution to

$$\hat{A}(\omega) \cdot \vec{u} = \vec{F}_0\delta(\vec{r}),$$

This is a challenging task because $\hat{G}$ depends on the specific realization of the quenched stress field, which plays the role of disorder. We would be content to compute $\overline{\hat{G}}$, the disorder-averaged Green's function. In our model, this cannot be done exactly. We employ the effective medium theory (EMT), a mean-field approximation that determines the optimal complex elastic moduli $\mu_E(\omega)$ and $\lambda_E(\omega)$ to represent the effect of scattering by disorder.

This task is similar to that faced in strongly interacting quantum systems, where one would like to compute the Green's function averaged over quantum fluctuations [40]. In that context, effective medium techniques have been developed under the name dynamical mean-field theory [41], which was shown to be exact in the limit of infinite dimensions, the mean-field limit [42].

We derive a self-consistent equation for the effective elastic modulus tensor $S_{ijkl}^E(\omega)$ under the effective medium approximation. We decompose the full elastic moduli into the effective moduli and the remainder $\Delta S_{ijkl} \equiv S_{ijkl} - S_{ijkl}^E(\omega)$. The key step is to choose $S_{ijkl}^E(\omega)$ such that the disorder-average of the true Green's function equals the EMT Green's function:

$$\overline{G_{ij}(\vec{r}; \vec{r}_0)} = G_{ij}^E(\vec{r} - \vec{r}_0), \tag{18}$$

where $G_{ij}^E(\vec{r} - \vec{r}_0)$ is the Green's function in terms of $S_{ijkl}^E(\omega)$. Here we assume that homogeneity and isotropy are restored in the effective Green's function, thus

$$G_{ij}^E(\vec{r}) = \sum_{\alpha=T,L} \int_q G_\alpha(q,\omega)\, P_{ij}^\alpha\, e^{i\vec{q}\cdot\vec{r}}, \tag{19}$$

where $G_T(q,\omega) = 1/(-\rho\omega^2 + \mu_E(\omega)q^2), G_L(q,\omega) = 1/(-\rho\omega^2 + \lambda_E(\omega)q^2)$, $P_{ij}^T = \delta_{ij} - \hat{q}_i\hat{q}_j, P_{ij}^L = \hat{q}_i\hat{q}_j$, and $\int_q = \int d^dq/(2\pi)^d$ over the region $q < \Lambda$. ( T and L stand for transverse and longitudinal, respectively. )

Let us note that the equation (18) cannot be imposed exactly, except in very simple models. One example of the latter is when a lattice has only a single bond with heterogeneity [16]. EMT is therefore a small-disorder (mean-field) approximation.

The EMT introduces one parameter, $v$, the correlation volume over which the EMT $\overline{G}_{ij}$ is attained; we take $v = 1/\int_q 1$ which is derived from comparison with previous results on spring networks [16]. This corresponds to a correlation volume on the order of the particle size.

The derivation of the EMT within the field theory formalism is sketched in Appendix 3.

We find that the final disorder-averaged Green's function depends on a random $d \times d$ matrix in the Gaussian orthogonal ensemble (GOE) [43]. Here we report results for the limit $\mu \ll 1$, for which $\lambda_E = 1 + \mathscr{O}(\mu)$: to leading order, the longitudinal Lamé modulus is not modified by the quench. Physically this limit means that the system is fragile like molecular glasses and weakly jammed materials, but it is adopted only for simplicity here. In this regime the key control parameter is

$$e = \left(v(d+2)^3 s_1\mu^4/4\right)^{-1}, \tag{20}$$

which measures the strength of moduli fluctuations. Introducing a fluctuating shear modulus $\mu_r(x) = \mu + e^{1/2}\mu x\bar{d}/d$ we find that $\mu_E$ satisfies

$$0 = \int dx \frac{\kappa(x)(\mu_E - \mu_r(x))}{1 - (1 - \frac{\mu_r(x)}{\mu_E})\frac{\bar{d}}{d}\left(1 + \rho\omega^2 v \int_q G_T(q,\omega)\right)}, \tag{21}$$

where $\kappa(x)$ is the convolution of a Gaussian with the spectral density of the GOE matrix. Note that the random variable $x$ can be interpreted as a normalized space-dependent shear modulus as discussed in Appendix 3. We emphasize that the GOE matrix whose spectrum appears in (21) is not put into the model, but emerges from its solution.

Remarkably, Eq.(21) *exactly* matches the form of an equation derived in [16], under the identifications $\mu_E \to k_\parallel$, an effective longitudinal stiffness; $\mu_r \to k_\alpha$, a fluctuating stiffness; and $z_0 = 2d^2/\bar{d}$, a lattice parameter. In [16], and in companion works [44, 45], the stiffnesses $k_\alpha$ were microscopic spring constants, whose distribution $\mathbb{P}(k_\alpha)$ was assumed to take simple tractable forms. In contrast, here we derive the relevant distribution $\kappa(x)$ from our model of quench dynamics.

## C. Wigner semicircle:

The simplest limit is $d \to \infty$, for which we expect EMT to be exact [41]. In $d = \infty$ the GOE spectrum is given by the Wigner semicircle law $\rho_W(x) = \sqrt{2d - x^2}/(\pi d)$ and $\kappa(x) = \rho(x)$ up to irrelevant corrections of relative order $1/d^2$. Using the fact that $\rho_W$ is supported on a finite interval $(-\sqrt{2d}, \sqrt{2d})$, we can determine the relevant scaling $e \sim 1/d$ and $x \sim 1$ for large $d$. Defining $\tilde{\omega} = \omega\sqrt{A_d/\mu}$ with $A_d = v\rho \int_q q^{-2}$, we take $\tilde{\omega} \sim 1$. Then we can derive a cubic equation for $y = \mu_E/\mu$:

$$0 = y^3 - y^2 + \frac{d}{2}e\left[y + \frac{A_d}{\mu}\omega^2\right], \tag{22}$$

The same equation has recently been derived for a lattice EMT, and analyzed in detail [46]. Translating these results, we find: (i) the solid is stable for $e < e_c = 1/(2d)$; (ii) near $e_c$ and at small $\omega$, $x$ satisfies a quadratic equation equivalent to that derived in [16], giving

$$\mu_E(\omega) = \tfrac{1}{2}\mu - i\sqrt{\frac{\mu A_d}{2}(\omega^2 - \omega_0^2)}, \tag{23}$$

where the onset frequency is

$$\omega_0 = \sqrt{\frac{\mu d}{A_d}(e_c - e)} \tag{24}$$

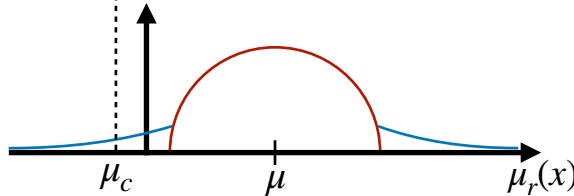

FIG. 2. Schematic distribution of fluctuating shear modulus $\mu_r$. In $d = \infty$, the distribution is a semi-circle; in finite dimensions, this develops a tail, which can lead to instability in large systems.

In this limit the vibrational properties are thus equivalent to those discussed in [16]. In particular, the density of vibrational states is $g(\omega) = (2\omega/\pi)\text{Im}\overline{G}_{ii}(0) \approx -(2\overline{d}\omega/\pi)\text{Im}[\mu_E]\int_q q^{-2}/|\mu_E|^2$, from which it follows that for $\omega > \omega_0$ Eq.(23) gives the non-Debye law $g(\omega) \propto \omega^2$ discussed in the introduction.

Scattering experiments measure the dynamic structure factor, from which one can extract, at each frequency $\omega$, a sound speed $\nu(\omega) = |\mu_E|/\text{Re}[\mu_E^{1/2}]$ [16] and the sound attentuation $\Gamma(\omega) \approx -\omega\text{Im}[\mu_E]/\text{Re}[\mu_E]$ [16]. In experiments [47, 48] and simulations [17] the sound speed is non-monotonic in frequency, showing a dip in the boson peak range where the sound attentuation crosses over from $\omega^{d+1}$ to $\omega^2$ behaviour; these features are reproduced by Eqs.(22),(23). Representative plots for these quantities in $d = 3$ are shown in Fig.3.

Quantitatively, we find $\Gamma(\omega) \approx (\pi d\mu e/4)v\rho\omega^2 g(\omega)/\text{Re}[\mu_E]$. In deriving Eq.(22), we ignored the hydrodynamic pole in the Green's function, which leads to the Debye law $g(\omega) \sim \omega^{d-1}$ for $\omega < \omega_0$. This thus leads to $\Gamma(\omega) \sim \omega^{d+1}$, which is Rayleigh scattering. Its amplitude is proportional to the variance of elastic moduli fluctuations, as observed [30].

### D. Finite-dimensional corrections:

GOE matrices have a spectrum whose bulk resembles the Wigner semicircle in all dimensions, with oscillatory corrections. Eq.(22) and its consequences can then be used in any dimension $d$ to determine the leading physics. However, for $d < \infty$, there is a new phenomenon completely absent in the Wigner semicircle: the spectrum develops an exponentially-decaying tail, as shown in Figure 2. This tail, which is Gaussian in $d = 2$ and $d = 3$, adds new excitations, which we expect to be localized. Formally, the tail extends to $\pm\infty$, implying that there are now *unstable* modes. Indeed, from Eq.(21), one can determine a stability condition $\mu_r > \mu_c = -\text{Re}[\mu_E]/\overline{d}$ [46]; the smallest modulus can be negative, but small, scaling as $1/d$. However, our results are derived for systems in the thermodynamic limit. Since $x$ corresponds to a spatially fluctuating modulus, in a system of $N$ particles there are approximately $N$ values of $x$ sampled from $\kappa(x)$. Define $x^*$ from $1/N = \int_{-\infty}^{x^*} dx\, \kappa(x)$. When $\mu_r(x^*) > \mu_c$, corresponding to small systems, the tail is irrelevant and the system is stable. In this case, using [49], we find that $\mu_E$ has a contribution $\delta\mu \sim -i\omega^2$ in the regime $0 < \omega < \omega_0$, which will lead to $g(\omega) \sim \omega^3$, on top of the Debye contribution. Instead, when $\mu_r(x^*) < \mu_c$, our quench has ended in an unstable state, and must further relax to a true inherent state. If $\kappa(x)$ is modified *ad-hoc* to vanish at $\mu_r(x_c) = \mu_c$ as $\kappa(x) \sim (x - x_c)^\beta$, then instead we find that $g(\omega) \sim \omega^{2\beta+1}$ [46].

These results are in qualitative agreement with previous findings, which indeed found a density of localized modes $g(\omega) \sim \omega^\alpha$ with $\alpha < 4$ in small systems [22]. In our model, the global stability of the system is controlled by the parameter $e$. However, we do not enforce *local stability* of the final state. Since $\alpha = 4$ is typically observed, these results imply that our assumption of an overdamped quench cannot be realistic in large systems. Future work should explicitly incorporate a condition of local mechanical stability in the IS, to properly predict the form of $\kappa(x)$ in realistic systems.

### III. CONCLUSION:

We propose a model for universal properties of amorphous solids based on the quench into an inherent state. Under a single universal distribution of quench stress, our model predicts (i) short-range correlations of elastic moduli, as observed [27, 29, 30]; (ii) long-range correlations of the IS stress, as observed [28, 31, 32]; (iii) exact reduction to previous models, shown to reproduce universal vibrational anomalies [16, 46]; and (iv) a tail of potentially unstable modes, beyond mean-field predictions, which leads to $g(\omega) \sim \omega^3$ in small systems and can rationalize larger exponents $g(\omega) \sim \omega^\alpha$ in large, stable systems.

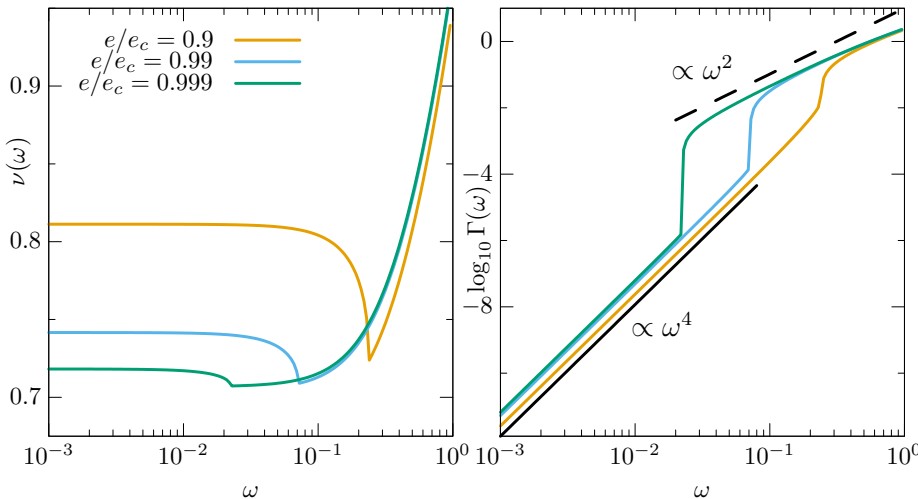

FIG. 3. (a) Sound speed $\nu(\omega)$ and (b) sound attenuation $\Gamma(\omega)$ for indicated values of $e/e_c$, in $d = 3$.

New results include the relationship between the elastic-moduli fluctuations and the stress correlations; explicit expressions for the stress correlations in arbitrary dimension; justification of mean-field models which formerly were derived from lattice models; and the inevitable extension of mean-field models to have an eigenvalue tail in finite dimensions.

Our model is based on an overdamped quench, which enforces mechanical equilibrium in the IS, but does not enforce stability. This allows unstable configurations, like a pencil standing on its head. At the global level, stability is ensured by choosing parameters such that all vibrational modes have a positive real frequency; in particular we should have $e < e_c$ as described above. However, we do not enforce stability *at each point in space.* In particular, the Hessian field $H_{ij}(\vec{r}, \vec{r}') = \partial^2 E/(\partial u_i(\vec{r}) \partial u_j(\vec{r}'))$ that controls local stability could have regions where it is not positive-definite. We predict Gaussian fluctuations of local elastic moduli, as observed in [27, 29], while it has very recently been argued that the moduli have a power-law tail due to localized modes [30]. We expect that these modes are created by local relaxation in regions of an unstable Hessian. To rigorously predict the density of small-frequency localized modes in large systems, and their potential modifications to elastic moduli fluctuations, future work should thus add local stability as a remaining feature to the model.

In addition, we have assumed for simplicity that the initial pre-quenched liquid state has homogenous short-time elastic constants. It would be useful and relevant to allow these to have short-range Gaussian fluctuations, which will be transformed under the quench and may alter the elastic properties of the glass.

**Acknowledgments:** We are grateful to E. Lerner and M. Wyart for comments on the manuscript, and to the referees for useful comments. MS thanks H. Mizuno for useful information on simulations and A. Ikeda for his encouragement. MS is supported by JSPS KAKENHI Grant No. 19J20036.

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

## IV.   APPENDIX 1. DERIVATION OF STRESS CORRELATIONS:

Here we show how to derive the distribution of the inherent stress field. In particular we show that our model predicts the observed long-range stress correlations, and moreover connects the magnitude of these correlations to the coefficients controlling vibrational properties. We use $\overline{\cdot}$ to denote a disorder average, not to be confused with the Grassmann-conjugation defined elsewhere.

The distribution of $\sigma$ is

$$
\begin{aligned}
\mathbb{P}[\sigma] &= \overline{\prod_{ij} \delta \left[\sigma_{ij} - \mathscr{P}_{ijkl}\tilde{\sigma}_{kl}\right]} \\
&= \int \mathscr{D}\hat{\tau} \, \overline{\exp \left\{ i \int_q \tau_{ij}(q) \left[\sigma_{ij}(q) - \mathscr{P}_{ijkl}(q)\tilde{\sigma}_{kl}(q)\right] \right\}} \\
&= \int \mathscr{D}\hat{\tau} \exp \left[ i \int_q \tau_{ij}(q)\sigma_{ij}(q) \right] \overline{\exp \left[ -i \int_q \tau_{ij}(q)\mathscr{P}_{ijkl}(q)\tilde{\sigma}_{kl}(q) \right]} \\
&= \int \mathscr{D}\hat{\tau} \exp \left[ i \int_q \tau_{ij}(q)\sigma_{ij}(q) \right] \exp \left[ - \int_q \tau_{ij}(q)\mathscr{P}'_{ijkl}(q)\tau_{kl}(-q) \right],
\end{aligned}
$$

where

$$
\begin{aligned}
\mathscr{P}'_{ijkl}(q) &= b_1 \left( \frac{2\mu}{1+2\mu} \right)^2 P^T{}_{ij}P^T{}_{kl} + \frac{b_2}{2}P^T{}_{ik}P^T{}_{jl} + \frac{b_2}{2}P^T{}_{il}P^T{}_{jk} + \frac{b_2}{(1+2\mu)^2}P^T{}_{ij}P^T{}_{kl} \\
&\equiv b'_1 P^T{}_{ij}P^T{}_{kl} + \frac{b_2}{2}\left(P^T{}_{ik}P^T{}_{jl} + P^T{}_{il}P^T{}_{jk}\right).
\end{aligned}
$$

and the inverse tensor $K^{-1}_{klmn} = b_1\delta_{kl}\delta_{mn} + b_2\delta_{km}\delta_{ln}$ is determined by the relation $K_{ijkl}K^{-1}_{klmn} = \delta_{im}\delta_{jn}$. Then we have

$$
b_1 = -\frac{1}{ds_1 s_2}\left(s_2 - \frac{s_1}{d}\right)
$$

$$
b_2 = \frac{1}{s_1}.
$$

Since $\mathscr{P}'_{ijkl}(q)q_i = \mathscr{P}'_{ijkl}(q)q_k = 0$, it is useful to perform a tensorial Helmholtz decomposition. We change the variable as $\tau_{ij}(q) = -iq_i\phi_j(q) - iq_j\phi_i(q) + \Psi_{ij}(q)$, where $q_i\Psi_{ij}(q) = q_j\Psi_{ij}(q) = 0$. Thus, we have

$$
\begin{aligned}
\mathbb{P}[\sigma] &= \int \mathscr{D}\phi\mathscr{D}\Psi \exp \left\{ i \int_q [-iq_i\phi_j(q) - iq_j\phi_i(q) + \Psi_{ij}(q)]\sigma_{ij}(q) - \int_q \Psi_{ij}(q)\mathscr{P}'_{ijkl}(q)\Psi_{kl}(-q) \right\} \\
&= \int \mathscr{D}\phi\mathscr{D}\Psi \exp \left\{ i \int_r \partial_i\phi_j(r) [\sigma_{ij}(r) + \sigma_{ji}(r)] - \int_q \left[\Psi_{ij}(q)\mathscr{P}'_{ijkl}(q)\Psi_{kl}(-q) - i\Psi_{ij}(q)\sigma_{ij}(q)\right] \right\}.
\end{aligned}
$$

Decomposing $\Psi$ and $\sigma$ as $\Psi = (\Psi + \Psi^T)/2 + (\Psi - \Psi^T)/2 = \Psi^S + \Psi^A$ and $\sigma = \sigma^S + \sigma^A$, we have

$$
\begin{aligned}
\mathbb{P}[\sigma] &= \delta\left[\sigma_{ij}(r) - \sigma_{ji}(r)\right]\delta\left[\partial_i\sigma_{ij}(r)\right] \int \mathscr{D}\Psi^S \exp \left\{ -\int_q \left[\Psi^S_{ij}(q)\mathscr{P}'_{ijkl}(q)\Psi^S_{kl}(-q) - i\Psi^S_{ij}(q)\sigma_{ij}(q)\right] \right\} \\
&\equiv \delta\left[\sigma_{ij}(r) - \sigma_{ji}(r)\right]\delta\left[\partial_i\sigma_{ij}(r)\right]\mathbb{P}^S[\sigma].
\end{aligned}
$$

The $\delta-$functions here are exactly the equations of mechanical equilibrium. The remaining symmetric part $\mathbb{P}^S[\sigma]$ further simplifies to

$$
\mathbb{P}^S[\sigma] = \int \mathscr{D}\Psi^S \exp \left\{ -\int_q \left[\Psi^S_{ij}(q)\left(b'_1\delta_{ij}\delta_{kl} + b_2\delta_{ik}\delta_{jl}\right)\Psi^S_{kl}(-q) - i\Psi^S_{ij}(q)\sigma_{ij}(q)\right] \right\},
$$

where we used $q_i\Psi^S_{ij}(q) = 0$.

In $d = 2$, setting $\Psi^S_{ij}(q) = -\epsilon_{ik}\epsilon_{jl}q_k q_l\psi(q)$, we have

$$
\begin{aligned}
\mathbb{P}^S[\sigma] &= \int \mathscr{D}\psi \exp \left\{ -\int_q [\epsilon_{im}\epsilon_{jn}\epsilon_{kp}\epsilon_{lq}\left(b'_1\delta_{ij}\delta_{kl} + b_2\delta_{ik}\delta_{jl}\right)q_m q_n q_p q_q\psi(q)\psi(-q) + i\epsilon_{ik}\epsilon_{jl}q_k q_l\sigma_{ij}(q)\psi(q)] \right\} \\
&= \int \mathscr{D}\psi \exp \left\{ -\int_q \left[(b'_1 + b_2)q^4\psi(q)\psi(-q) + i\epsilon_{ik}\epsilon_{jl}q_k q_l\sigma_{ij}(q)\psi(q)\right] \right\}.
\end{aligned}
$$

To match the notation of [37, 38], we set $1/2\tilde{\eta} = b_1' + b_2$. Thus,

$$
\begin{aligned}
\mathbb{P}^S[\sigma] &= \int \mathscr{D}\psi \exp\left\{-\frac{\tilde{\eta}^{-1}}{2}\int_q q^4 \left[\psi(q) + i\epsilon_{ik}\epsilon_{jl}\hat{q}_k\hat{q}_l q^{-2}\tilde{\eta}\sigma_{ij}(q)\right]\left[\psi(-q) + i\epsilon_{mp}\epsilon_{nq}\hat{q}_p\hat{q}_q q^{-2}\tilde{\eta}\sigma_{mn}(-q)\right]\right. \\
&\qquad \left. -\frac{\tilde{\eta}}{2}\int_q \epsilon_{ik}\epsilon_{jl}\hat{q}_k\hat{q}_l\sigma_{ij}(q)\epsilon_{mp}\epsilon_{nq}\hat{q}_p\hat{q}_q\sigma_{mn}(-q)\right\} \\
&= \frac{1}{Z_\sigma}\exp\left[-\frac{\tilde{\eta}}{2}\int_q \epsilon_{ik}\epsilon_{jl}\hat{q}_k\hat{q}_l\sigma_{ij}(q)\epsilon_{mp}\epsilon_{nq}\hat{q}_p\hat{q}_q\sigma_{mn}(-q)\right] \\
&= \frac{1}{Z_\sigma}\exp\left[-\frac{\tilde{\eta}}{2}\int_r \mathrm{tr}^2\sigma(r)\right],
\end{aligned}
$$

where $Z_\sigma$ is the normalization constant and we used $q_i\sigma_{ij}(q) = 0$. The coefficient $\tilde{\eta}$ is given by

$$
\tilde{\eta} = \frac{(1+2\mu)^2}{2\{4\mu^2 b_1 + [1 + (1+2\mu)^2]b_2\}}.
$$

In $d = 3$ the computations are similar. The relevant tensorial Helmholtz decomposition is discussed in [37].

## V. APPENDIX 2. DERIVATION OF ELASTIC MODULI FLUCTUATIONS:

### 1. Bulk modulus

Here, we derive fluctuations in bulk modulus. The fluctuating part of $S_{ijkl}$ reads

$$
\delta S_{ijkl} = (\mathscr{S}_{ijklmn} + \mathscr{P}_{ikmn}\delta_{jl})\tilde{\sigma}_{mn}.
$$

When we consider a homogeneous bulk deformation $\varepsilon_{kl} = \varepsilon\delta_{kl}/d$, the pressure fluctuation is

$$
\delta p = \frac{1}{d}\delta S_{iikl}\varepsilon_{kl} = \frac{1}{d^2}\delta S_{iikk}\varepsilon.
$$

Thus, the bulk modulus fluctuation is

$$
\begin{aligned}
\delta K = \delta p/\varepsilon &= \frac{1}{d^2}(\mathscr{S}_{iikkmn} + \mathscr{P}_{ikmn}\delta_{ik})\tilde{\sigma}_{mn} \\
&= -\frac{1}{d^2}\left\{[4 + 5c(d-1)]\hat{q}_m\hat{q}_n + P^T{}_{mn}\right\}\tilde{\sigma}_{mn} \equiv \mathscr{K}_{mn}\tilde{\sigma}_{mn}.
\end{aligned}
$$

The probability distribution function of $\delta K$ can be written as

$$
\begin{aligned}
\mathbb{P}[\delta K] &= \overline{\delta[\delta K - \mathscr{K} : \hat{\tilde{\sigma}}]} \\
&= \overline{\int \mathscr{D}\tau \exp\left\{i\int_q \tau(-q)[\delta K(q) - \hat{\mathscr{K}}(q) : \hat{\tilde{\sigma}}(q)]\right\}} \\
&= \int \mathscr{D}\tau \exp\left[i\int_q \tau(-q)\delta K(q)\right]\exp\left\{-2\int_q \tau(q)\tau(-q)\hat{\mathscr{K}}(q) : K^{-1} : \hat{\mathscr{K}}(-q)\right\}.
\end{aligned}
$$

Using the definition of $\hat{\mathscr{K}}$, we have

$$
\mathscr{K}_{ij}(q)K^{-1}_{ijkl}\mathscr{K}_{kl}(-q) = \frac{1}{d^4}\left\{b_1[4 + (5c+1)(d-1)]^2 + b_2[4 + 5c(d-1)]^2 + b_2(d-1)\right\} \equiv \frac{K_{\mu,d}}{4\beta}.
$$

Finally, we obtain

$$
\mathbb{P}[\delta K] = \frac{1}{Z_K}\exp\left[-\frac{\beta}{2K_{\mu,d}}\int_r \delta K(r)^2\right].
$$

Thus, the correlation of the bulk modulus is

$$
\langle \delta K(r)\delta K(0)\rangle = \frac{1}{2\beta}K_{\mu,d}\delta(r).
$$

This is always short range.

## 2. Shear modulus

The shear modulus fluctuation is

$$\delta\mu = \frac{1}{(d+2)(d-1)}\left(\mathscr{S}_{ijijmn} - \frac{1}{d}\mathscr{S}_{iikkmn} + \mathscr{P}_{iimn}\delta_{jj} - \frac{1}{d}\mathscr{P}_{ikmn}\delta_{ik}\right)\tilde{\sigma}_{mn} \equiv \mathscr{M}_{mn}\tilde{\sigma}_{mn}, \tag{25}$$

where

$$(d+2)d(d-1)\mathscr{M}_{mn} = -[(d-1)(d^2-2d-5)c + 2(d^2+d-2)]\hat{q}_m\hat{q}_n + (d^2+1)P^T{}_{mn}.$$

As in the case of $\delta K$, the probability distribution of $\delta\mu$ is

$$\mathbb{P}[\delta\mu] = \overline{\delta(\delta\mu - \mathscr{M} : \hat{\tilde{\sigma}})}$$

$$= \int \mathscr{D}\tau \exp\left[i\int_q \tau(-q)\delta\mu(q)\right]\exp\left[-2\int_q \tau(q)\tau(-q)\mathscr{M}(q) : K^{-1} : \mathscr{M}(-q)\right].$$

Performing the contractions of $\mathscr{M}$, we have

$$\beta(d+2)^2 d^2(d-1)^2 \mathscr{M}(q) : K^{-1} : \mathscr{M}(-q)$$
$$= \beta b_1\{[(d-1)(d^2-2d-5)c + 2(d^2+d-2)] + (d^2+1)(d-1)\}^2$$
$$\quad + \beta b_2[(d-1)(d^2-2d-5)c + 2(d^2+d-2)]^2 + \beta b_2(d^2+1)^2(d-1)$$
$$\equiv \frac{1}{4}(d+2)^2 d^2(d-1)^2 M_{\mu,d}.$$

Thus,

$$\mathbb{P}[\delta\mu] = \frac{1}{Z_M}\exp\left[-\frac{\beta}{2M_{\mu,d}}\int_r \delta\mu(r)^2\right].$$

Thus, the correlation function is

$$\langle\delta\mu(r)\delta\mu(0)\rangle = \frac{1}{2\beta}M_{\mu,d}\delta(r).$$

The results of $\delta K$ and $\delta\mu$ are consistent with the numerical observation of ordinary glasses, where indeed these moduli have short-range correlations.

## VI.   APPENDIX 3. DERIVATION OF EFFECTIVE MEDIUM THEORY

Here we derive the effective medium theory. We will eventually need the following relations:

$$\begin{cases} V_{efmn} & = V_{femn} = V_{efnm} = V_{mnef} \\ V_{eemn} & = 2(1-c)\hat{q}_m\hat{q}_n \\ V_{efmm} & = V_{efmn}\hat{q}_m\hat{q}_n = 2(1-c)\hat{q}_e\hat{q}_f \end{cases}. \tag{26}$$

We have

$$\left[\hat{A}^E(\omega) + \Delta\hat{A}(\omega)\right]\cdot\vec{u} = \vec{F}_0\delta(\vec{r}), \tag{27}$$

where $\hat{A}^E_{il}(\omega) = -\rho\omega^2\delta_{il} - S^E_{ijkl}\partial_j\partial_k$ and $\Delta A_{il}(\omega) = -\partial_j\Delta S_{ijkl}\partial_k$. In Fourier space, the left hand side of (27) is

$$\left[A^E_{il}(\omega) + \Delta A_{il}(\omega)\right]\int_q e^{i\vec{q}\cdot\vec{r}}u_l(\vec{q})$$

$$= \int_q e^{i\vec{q}\cdot\vec{r}}\left[(-\rho\omega^2\delta_{il} + S^E_{ijkl}q_jq_k) + [q_kq_j\Delta S_{ijkl} - iq_k(\partial_j\Delta S_{ijkl})]\right]u_l(\vec{q})$$

$$\equiv \int_q e^{i\vec{q}\cdot\vec{r}}\left[A^E_{il}(\vec{q}) + \Delta A_{il}(\vec{q};\vec{r})\right]u_l(\vec{q})$$

Then, (27) is written as

$$\vec{u}(\vec{q}) = \hat{G}^E(\vec{q}) \cdot \vec{F}_0 - \int_{r,q'} e^{-i(\vec{q}-\vec{q}')\cdot\vec{r}} \hat{G}^E(\vec{q}) \cdot \Delta\hat{A}(\vec{q}';\vec{r}) \cdot \vec{u}(\vec{q}'),$$

where $\hat{G}^E(\vec{q}) = \hat{A}^E(\vec{q})^{-1}$ is the effective disorder-averaged Green's function. Solving this equation by iteration, we obtain

$$\vec{u}(\vec{q}_1)$$

$$= \hat{G}^E(\vec{q}_1) \cdot \vec{F}_0 + \int_{r_1,q_2} e^{-i(\vec{q}_1-\vec{q}_2)\cdot\vec{r}_1} \hat{G}^E(\vec{q}_1) \cdot \left[-\Delta\hat{A}(\vec{q}_2;\vec{r}_1)\right] \cdot (\hat{G}^E(\vec{q}_2)) \cdot \vec{F}_0$$

$$+ \int_{r_1,r_2,q_2,q_3} e^{-i(\vec{q}_1-\vec{q}_2)\cdot\vec{r}_1 - i(\vec{q}_2-\vec{q}_3)\cdot\vec{r}_2} \hat{G}^E(\vec{q}_1) \cdot \left[-\Delta\hat{A}(\vec{q}_2;\vec{r}_1)\right] \cdot \hat{G}^E(\vec{q}_2) \cdot \left[-\Delta\hat{A}(\vec{q}_3;\vec{r}_2)\right] \cdot \hat{G}^E(\vec{q}_3) \cdot \vec{F}_0 + \cdots$$

$$= \hat{G}^E(\vec{q}_1) \cdot \vec{F}_0$$

$$+ \sum_{n=1}^{\infty} \int_{\substack{r_1,\cdots,r_n \\ q_2,\cdots,q_{n+1}}} e^{-i(\vec{q}_1-\vec{q}_2)\cdot\vec{r}_1 - \cdots - i(\vec{q}_n-\vec{q}_{n+1})\cdot\vec{r}_n} \hat{G}^E(\vec{q}_1) \cdot \left[\prod_{m=1}^{n} \left[-\Delta\hat{A}(\vec{q}_{m+1};\vec{r}_m)\right] \cdot \hat{G}^E(\vec{q}_{m+1})\right] \cdot \vec{F}_0$$

$$\equiv \hat{G}^E(\vec{q}_1) \cdot \vec{F}_0 + \sum_{n=1}^{\infty} \hat{\mathscr{A}}_n \cdot \vec{F}_0. \tag{28}$$

So far our manipulations are exact. The term of order $n$ in this expansion is a contribution to response from scattering across disorder at $n$ sites $\vec{r}_1, \ldots, \vec{r}_n$. To motivate the coherent potential approximation, consider the case when there is only a single defect in the medium: $\Delta\hat{A}(\vec{q}_{m+1};\vec{r}_m) = \tilde{\hat{A}}(\vec{q}_{m+1})\delta(\vec{r}_m - \vec{r}_0)$. In that case all the $\vec{r}_i$ integrals can be done immediately and we have

$$\hat{\mathscr{A}}_n = \int_{q_2,\cdots,q_{n+1}} e^{-i(\vec{q}_1-\vec{q}_{n+1})\cdot\vec{r}_0} \hat{G}^E(\vec{q}_1) \cdot \left[\prod_{m=1}^{n} \left[-\tilde{\hat{A}}(\vec{q}_{m+1})\right] \cdot \hat{G}^E(\vec{q}_{m+1})\right]$$

$$= -\int_{q_{n+1}} e^{-i(\vec{q}_1-\vec{q}_{n+1})\cdot\vec{r}_0} \hat{G}^E(\vec{q}_1) \cdot \left[-\int_k \tilde{\hat{A}}(\vec{k}) \cdot \hat{G}^E(\vec{k})\right]^{n-1} \cdot \tilde{\hat{A}}(\vec{q}_{n+1}) \cdot \hat{G}^E(\vec{q}_{n+1}) \qquad \text{single-defect}$$

This leads to a contribution to the response from scattering $n$ times off the defect.

Now, for a general $\Delta\hat{A}$, there are also contributions from scattering off multiple defects. In the coherent potential approximation, these are neglected. We first write, exactly,

$$\hat{\mathscr{A}}_n = \int_{\substack{r_1,\cdots,r_n \\ q_2,\cdots,q_{n+1}}} e^{-i\sum_{m=1}^{n}(\vec{q}_m-\vec{q}_{m+1})\cdot\vec{r}_m} v^{n-1} \prod_{m=1}^{n-1} \left[v^{-1} - \delta(\vec{r}_m - \vec{r}_{m+1}) + \delta(\vec{r}_m - \vec{r}_{m+1})\right]$$

$$\hat{G}^E(\vec{q}_1) \cdot \left[\prod_{m=1}^{n} \left[-\Delta\hat{A}(\vec{q}_{m+1};\vec{r}_m)\right] \cdot \hat{G}^E(\vec{q}_{m+1})\right]$$

A constant $v$, with units of volume, has been inserted. If $v$ is the correlation volume of the disorder, then the term $v^{-1} - \delta(\vec{r}_m - \vec{r}_{m+1})$ will average to zero, typically, since when one of the position coordinates is integrated out, the $\delta-$function equates the disorder between the two defects, and the first term, by definition, picks up a contribution of the correlation volume. Thus the CPA is generated by neglecting the terms $v^{-1} - \delta(\vec{r}_m - \vec{r}_{m+1})$. The above procedure mimics the T-matrix construction used in lattice models [16, 54, 55].

Then

$$\hat{\mathscr{A}}_n = \int_{\substack{r_1,\cdots,r_n \\ q_2,\cdots,q_{n+1}}} e^{-i\sum_{m=1}^{n}(\vec{q}_m-\vec{q}_{m+1})\cdot\vec{r}_m} v^{n-1} \prod_{m=1}^{n-1} \delta(\vec{r}_m - \vec{r}_{m+1})$$

$$\hat{G}^E(\vec{q}_1) \cdot \left[\prod_{m=1}^{n} \left[-\Delta\hat{A}(\vec{q}_{m+1};\vec{r}_m)\right] \cdot \hat{G}^E(\vec{q}_{m+1})\right] + \cdots$$

$$= \int_{r_1,q_2,\cdots,q_{n+1}} e^{-i(\vec{q}_1-\vec{q}_{n+1})\cdot\vec{r}_1} v^{n-1} \hat{G}^E(\vec{q}_1) \cdot \left[\prod_{m=1}^{n} \left[-\Delta\hat{A}(\vec{q}_{m+1};\vec{r}_1)\right] \cdot \hat{G}^E(\vec{q}_{m+1})\right] + \cdots$$

$$= -\int_{r,q'} e^{-i(\vec{q}_1-\vec{q}')\cdot\vec{r}} \hat{G}^E(\vec{q}_1) \cdot \left[-v\int_k \Delta\hat{A}(\vec{k};\vec{r}) \cdot \hat{G}^E(\vec{k})\right]^{n-1} \cdot \Delta\hat{A}(\vec{q}';\vec{r}) \cdot \hat{G}^E(\vec{q}') + \cdots,$$

This expression is seen to be a generalization of the single-defect form above. When we neglect higher order terms, (28) simplifies to

$$\vec{u}(\vec{q}) \simeq \hat{G}^E(\vec{q}) \cdot \left[ 1 - \int_{r,q'} e^{-i(\vec{q}-\vec{q}\,')\cdot\vec{r}} \sum_{n=1}^{\infty} \left[ -v \int_k \Delta\hat{A}(\vec{k};\vec{r}) \cdot \hat{G}^E(\vec{k}) \right]^{n-1} \cdot \Delta\hat{A}(\vec{q}\,';\vec{r}) \cdot \hat{G}^E(\vec{q}\,') \right] \cdot \vec{F}_0$$

$$= \hat{G}^E(\vec{q}) \cdot \left[ 1 - \int_{r,q'} e^{-i(\vec{q}-\vec{q}\,')\cdot\vec{r}} \left[ 1 + v \int_k \Delta\hat{A}(\vec{k};\vec{r}) \cdot \hat{G}^E(\vec{k}) \right]^{-1} \cdot \Delta\hat{A}(\vec{q}\,';\vec{r}) \cdot \hat{G}^E(\vec{q}\,') \right] \cdot \vec{F}_0.$$

Thus, the total Green's function is written as

$$\hat{G}(\vec{q}) = \hat{G}^E(\vec{q}) - \hat{G}^E(\vec{q}) \cdot \int_{r,q'} e^{-i(\vec{q}-\vec{q}\,')\cdot\vec{r}} \left[ 1 + v \int_k \Delta\hat{A}(\vec{k};\vec{r}) \cdot \hat{G}^E(\vec{k}) \right]^{-1} \cdot \Delta\hat{A}(\vec{q}\,';\vec{r}) \cdot \hat{G}^E(\vec{q}\,'). \qquad (29)$$

When we take the disorder average, $\hat{A}(\vec{k};\vec{r})$ and $\hat{A}(\vec{q}\,';\vec{r})$ in the second term no longer depend on $\vec{r}$. Thus,

$$\overline{\hat{G}(\vec{q})} = \hat{G}^E(\vec{q}) - \hat{G}^E(\vec{q}) \cdot \int_{r,q'} e^{-i(\vec{q}-\vec{q}\,')\cdot\vec{r}} \overline{\left[ 1 + v \int_k \Delta\hat{A}(\vec{k};\vec{r}) \cdot \hat{G}^E(\vec{k}) \right]^{-1} \cdot \Delta\hat{A}(\vec{q}\,';\vec{r}) \cdot \hat{G}^E(\vec{q}\,')} \qquad (30)$$

$$= \hat{G}^E(\vec{q}) - \hat{G}^E(\vec{q}) \cdot \overline{\left[ 1 + v \int_k \Delta\hat{A}(\vec{k};\vec{r}) \cdot \hat{G}^E(\vec{k}) \right]^{-1} \cdot \Delta\hat{A}(\vec{q};\vec{r}) \cdot \hat{G}^E(\vec{q})}. \qquad (31)$$

When we assume that the disorder average of the total Green's function can be written as $\overline{\hat{G}(\vec{q})} = G_T(q)P^T + G_L(q)\hat{q}\hat{q}$, we have

$$(d-1)G_T(q) = (d-1)G_T^E(q) - G_T^E(\vec{q})^2 \mathrm{tr}\left\{ \overline{\left[ 1 + v \int_k \Delta\hat{A}(\vec{k};\vec{r}) \cdot \hat{G}^E(\vec{k}) \right]^{-1} \cdot \Delta\hat{A}(\vec{q};\vec{r}) \cdot P^T} \right\}$$

and

$$G_L(q) = G_L^E(q) - G_L^E(\vec{q})^2 \mathrm{tr}\left\{ \overline{\left[ 1 + v \int_k \Delta\hat{A}(\vec{k};\vec{r}) \cdot \hat{G}^E(\vec{k}) \right]^{-1} \cdot \Delta\hat{A}(\vec{q};\vec{r}) \cdot \hat{q}\hat{q}} \right\}.$$

We assume that the disorder average of the total Green's function is equal to $\hat{G}^E(\vec{r})$ under the effective medium approximation. Thus, we have two equations:

$$\mathrm{tr}\left\{ \overline{\left[ 1 + v \int_k \Delta\hat{A}(\vec{k};\vec{r}) \cdot \hat{G}^E(\vec{k}) \right]^{-1} \cdot \Delta\hat{A}(\vec{q};\vec{r}) \cdot P^T} \right\} = 0 \qquad (32)$$

$$\mathrm{tr}\left\{ \overline{\left[ 1 + v \int_k \Delta\hat{A}(\vec{k};\vec{r}) \cdot \hat{G}^E(\vec{k}) \right]^{-1} \cdot \Delta\hat{A}(\vec{q};\vec{r}) \cdot \hat{q}\hat{q}} \right\} = 0 \qquad (33)$$

To continue, we use an identity for a matrix X with a parameter $z$

$$\frac{d}{dz} \log \det X = \mathrm{tr}\left[ X^{-1} \cdot \frac{d}{dz}X \right].$$

From this, we have

$$\frac{\delta}{\delta J_\alpha(q)} \log \det \left[ 1 + v \int_k \Delta\hat{A}(\vec{k};\vec{r}\,') \cdot \left[(G_\alpha^E + J_\alpha)P^\alpha\right] \right]\bigg|_{J_\alpha=0}$$

$$= v\,\mathrm{tr}\left\{ \left[ 1 + v \int_k \Delta\hat{A}(\vec{k};\vec{r}\,') \cdot G_\alpha^E(\vec{k})P^\alpha \right]^{-1} \cdot \Delta\hat{A}(\vec{q};\vec{r}) \cdot P^\alpha \right\},$$

where $\alpha = T$ or $L$, and $\hat{P}^L = \hat{q}\hat{q}$. Using this identity, (32) and (33) can be expressed by a single equation

$$0 = \overline{\frac{\delta}{\delta J_\alpha(q)} \log \det \left[ 1 + v \int_k \Delta\hat{A}(\vec{k};\vec{r}) \cdot (G_\alpha^E + J_\alpha)P^\alpha \right]}\bigg|_{\hat{J}=0}$$

$$= \overline{\frac{\delta}{\delta J_\alpha(q)} \lim_{n\to 0} \frac{1}{n} \left\{ \det{}^n \left[ 1 + v \int_k \Delta\hat{A}(\vec{k};\vec{r}) \cdot (G_\alpha^E + J_\alpha)P^\alpha \right] - 1 \right\}}\bigg|_{\hat{J}=0}$$

$$= \overline{\frac{\delta}{\delta J_\alpha(q)} \lim_{n\to 0} \frac{1}{n} \det{}^n \left[ 1 + v \int_k \Delta\hat{A}(\vec{k};\vec{r}) \cdot (G_\alpha^E + J_\alpha)P^\alpha \right]}\bigg|_{\hat{J}=0}. \qquad (34)$$

To express this determinant by a Gaussian integral, we will use anticommuting Grassmann variables, satisfying

$$\theta_i\theta_j = -\theta_j\theta_i$$

All our notations and definitions follow [56]. We introduce a second set of variables $\bar{\theta}_i$, anticommuting with the $\theta_i$, but independent. The key identity is

$$\det M = \int \prod_i d\theta_i d\bar{\theta}_i e^{M_{ij}\bar{\theta}_i\theta_j}.$$

Importantly, these identities hold for arbitrary non-singular $M$. Using this identity, the generating functional in (34) can be rewritten as

$$W_n[J] = \det{}^n\left[1 + v\int_k \Delta\hat{A}(\vec{q};\vec{r})\cdot(G_\alpha^E + J_\alpha)P^\alpha\right]$$

$$= \int \mathscr{D}\theta\mathscr{D}\bar{\theta}\exp\left\{\left[1 + v\int_k \Delta\hat{A}(\vec{q};\vec{r})\cdot(G_\alpha^E + J_\alpha)P^\alpha\right]_{ij}\bar{\theta}_i^a\theta_j^a\right\}, \tag{35}$$

where $a = 1,\ldots,n$ is the replica index.

### 1. Evaluation of $W_n[J]$

$\Delta\hat{A}(\vec{q};\vec{r})$ is written as

$$\Delta A_{il}(\vec{q};\vec{r}) = -S_{ijkl}^E q_k q_j + (q_k q_j - iq_k\partial_j)S_{ijkl}$$

$$= \rho\omega^2\delta_{il} - S_{ijkl}^E q_k q_j - \rho\omega^2\delta_{il} + \int_{q'} e^{i\vec{q}'\cdot\vec{r}}(q_j + q_j')q_k S_{ijkl}(\vec{q}')$$

$$= -A_{il}^E(\vec{q}) - \rho\omega^2\delta_{il} + \int_{q'} e^{i\vec{q}'\cdot\vec{r}}(q_j + q_j')q_k S_{ijkl}(\vec{q}').$$

Thus, the exponent in (35) is

$$\left[1 + v\int_q \Delta\hat{A}(\vec{q};\vec{r})\cdot(G_\alpha^E + J_\alpha)P^\alpha\right]_{ij}$$

$$= \delta_{ij} - v\int_q \left[A_{il}^E(\vec{q}) + \rho\omega^2\delta_{il}\right](G_\alpha^E + J_\alpha)P_{lj}^\alpha(\vec{q}) + v\int_{q,q'} e^{i\vec{q}'\cdot\vec{r}}(q_m + q_m')q_k S_{imkl}(\vec{q}')(G_\alpha^E + J_\alpha)P_{lj}^\alpha(\vec{q})$$

$$= \delta_{ij}\left[1 - v\left(1 - \frac{1}{d}\right)\mu^E I_T - \frac{v}{d}(\lambda^E + 2\mu^E)I_L\right]$$

$$+ v\int_q q_k q_l\left[(G_T^E + J_T)P^T{}_{mj}(\vec{q}) + (G_L^E + J_L)\hat{q}_m\hat{q}_j\right]\int_{q'} e^{i\vec{q}'\cdot\vec{r}}S_{iklm}(\vec{q}')$$

$$= \delta_{ij}\left[1 - v\left(1 - \frac{1}{d}\right)\mu^E I_T - \frac{v}{d}(\lambda^E + 2\mu^E)I_L\right]$$

$$+ v\left[\frac{1}{d}\delta_{kl}\delta_{mj}I_T + \int_q q^2(G_L^E + J_L - G_T^E - J_T)\hat{q}_k\hat{q}_l\hat{q}_m\hat{q}_j\right]\int_{q'} e^{i\vec{q}'\cdot\vec{r}}S_{iklm}(\vec{q}'), \tag{36}$$

where $I_\alpha = \int_q q^2(G_\alpha^E + J_\alpha)$. We used the fact that $G_\alpha^E(q)$ and $J_\alpha(q)$ only depend on $q = |\vec{q}|$ and $\int_q f(q)\hat{q}_i\hat{q}_j = \delta_{ij}\int_q f(q)/d$, where $f(q) = G_\alpha^E(q)$ or $J_\alpha(q)$. We can rewrite the integral

$$\int_q q^2(G_L^E + J_L - G_T^E - J_T)\hat{q}_k\hat{q}_l\hat{q}_m\hat{q}_j = a(\delta_{kl}\delta_{mj} + \delta_{km}\delta_{lj} + \delta_{lm}\delta_{kj}),$$

since the integrands depend on $|\vec{q}|$ only, and the tensorial part is completely symmetric. But by taking a contraction, we see

$$\int_q q^2(G_L^E + J_L - G_T^E - J_T)\hat{q}_i\hat{q}_i\hat{q}_j\hat{q}_j = a(d^2 + d + d),$$

which fixes the value of $a$. Then (36) becomes

$$\delta_{ij}\left[1 - v\left(1 - \frac{1}{d}\right)\mu^E I_T - \frac{v}{d}(\lambda^E + 2\mu^E)I_L\right] + \frac{v}{d(d+2)}[A\delta_{kl}\delta_{mj} + B(\delta_{km}\delta_{lj} + \delta_{lm}\delta_{kj})]\int_{q'}e^{i\vec{q}\,'\cdot\vec{r}}S_{iklm}(\vec{q}\,')$$

$$= \delta_{ij}\tilde{C} + \frac{v}{d(d+2)}\int_{q'}e^{i\vec{q}\,'\cdot\vec{r}}[A(\mathscr{S}_{ikkjmn} + \mathscr{P}_{ikmn}\delta_{kj}) + B(\mathscr{S}_{ikjkmn} + \mathscr{P}_{ijmn}\delta_{kk} + \mathscr{S}_{ijkkmn} + \mathscr{P}_{ikmn}\delta_{jk})]\overline{\sigma}_{mn}, \quad (37)$$

where $A = (d+1)I_T + I_L$, $B = I_L - I_T$, and

$$\tilde{C} = 1 + v\left(1 - \frac{1}{d}\right)(\mu - \mu^E)I_T + \frac{v}{d}(1 + 2\mu - \lambda^E - 2\mu^E)I_L.$$

We next need to perform the contractions of $\mathscr{S}$ and $\mathscr{P}$ in (37). To this end, we simplify (8) and (9). Using (26), we have

$$P_{jm}^T P_{ln}^T + P_{jn}^T P_{lm}^T - 2cP_{jl}^T\hat{q}_m\hat{q}_n = -V_{jlmn} + \delta_{jm}\delta_{ln} + \delta_{jn}\delta_{lm} - 2c\delta_{jl}\hat{q}_m\hat{q}_n.$$

Thus, the tensor $\mathscr{S} + \mathscr{P}\delta$ simplifies to

$$\mathscr{S}_{ijklmn} + \mathscr{P}_{ikmn}\delta_{jl}$$
$$= -\frac{1}{2(1-c)}[2c(\delta_{ij}\delta_{ke}\delta_{lf} + \delta_{kl}\delta_{ie}\delta_{jf}) + (1-c)(2\delta_{ik}\delta_{je}\delta_{lf} + \delta_{jl}\delta_{ie}\delta_{kf} + \delta_{il}\delta_{je}\delta_{kf} + \delta_{jk}\delta_{ie}\delta_{lf})]V_{efmn}$$
$$+ \tfrac{1}{2}\delta_{ik}(\delta_{jm}\delta_{ln} + \delta_{jn}\delta_{lm} - 2c\delta_{jl}\hat{q}_m\hat{q}_n).$$

Using this, the contractions in (37) are evaluated as follows:

$$\mathscr{S}_{ikkjmn} + \mathscr{P}_{ikmn}\delta_{kj} = -\tfrac{1}{2}(d + 2c_r - 1)V_{ijmn} + \tfrac{1}{2}(\delta_{im}\delta_{jn} + \delta_{in}\delta_{jm} - 2\delta_{ij}\hat{q}_m\hat{q}_n), \quad (38)$$

$$\mathscr{S}_{ikjkmn} + \mathscr{P}_{ijmn}\delta_{kk} = -\tfrac{1}{2}(d + 2c_r - 2)V_{ijmn} + \delta_{ij}[\delta_{mn} - (cd + 2(1-c))\hat{q}_m\hat{q}_n]. \quad (39)$$

$$\mathscr{S}_{ijkkmn} + \mathscr{P}_{ikmn}\delta_{jk} = -\tfrac{1}{2}\left((c_r - 2)d + 5\right)V_{ijmn} + \tfrac{1}{2}(\delta_{jm}\delta_{in} + \delta_{jn}\delta_{im} - 6c\delta_{ij}\hat{q}_m\hat{q}_n). \quad (40)$$

For later convenience, we further consider two contractions here. The first one is

$$A(\mathscr{S}_{ikkjmm} + \mathscr{P}_{ikmm}\delta_{kj}) + B(\mathscr{S}_{ikjkmm} + \mathscr{P}_{ijmm}\delta_{kk} + \mathscr{S}_{ijkkmm} + \mathscr{P}_{ikmm}\delta_{jk})$$
$$= -\frac{2}{c_r}\{[d^2 + (c_r + 1)d - 4]I_T + (c_r d + 4c_r + 2)I_L\}\hat{q}_i\hat{q}_j + \frac{2}{c_r}(d - c_r + 1)(-I_T + I_L)\delta_{ij}. \quad (41)$$

The second one is

$$V_{ijmn}V_{klmn} = 2V_{ijkl} - 4c(1-c)\hat{q}_i\hat{q}_j\hat{q}_k\hat{q}_l. \quad (42)$$

### 2. Disorder average

From the results of the previous sections, we can write

$$W_n[J] = \int\mathscr{D}\theta\mathscr{D}\overline{\theta}e^{\tilde{C}\overline{\theta}_i^a\theta_i^a}\exp\left\{\int_q e^{i\vec{q}\cdot\vec{r}}\mathscr{X}_{mn}\tilde{\sigma}_{mn}\right\}$$

where

$$\mathscr{X}_{mn} = v[A(\mathscr{S}_{ikkjmn} + \mathscr{P}_{ikmn}\delta_{kj}) + B(\mathscr{S}_{ikjkmn} + \mathscr{P}_{ijmn}\delta_{kk} + \mathscr{S}_{ijkkmn} + \mathscr{P}_{ikmn}\delta_{jk})]\overline{\theta}_i^a\theta_j^a/d(d+2). \quad (43)$$

Averaging out $\tilde{\sigma}$, we obtain

$$\overline{W_n[J]} = \int\mathscr{D}\theta\mathscr{D}\overline{\theta}e^{\tilde{C}\overline{\theta}_i^a\theta_i^a}\exp\left[\int_q\mathscr{X}(\vec{q}) : K^{-1} : \mathscr{X}(-\vec{q})\right]$$

$$= \int\mathscr{D}\theta\mathscr{D}\overline{\theta}e^{\tilde{C}\overline{\theta}_i^a\theta_i^a}\exp\left[\int_q b_1(\mathscr{X}_{mm})^2 + b_2(\mathscr{X}^2)_{mm}\right]. \quad (44)$$

Using (41), the term proportional to $b_1$ is

$$b_1 \int_q (\mathscr{X}_{mm})^2 = \frac{b_1 v}{d^3(d+2)^3 c_r^2} \overline{\theta}_i^a \theta_j^a \overline{\theta}_k^b \theta_l^b [P(\delta_{ik}\delta_{jl} + \delta_{il}\delta_{jk}) + Q\delta_{ij}\delta_{kl}],$$

where

$$P = 4\{[d^2 + (c_r+1)d - 4]I_T + (c_r d + 4c_r + 2)I_L\}^2 \tag{45}$$

$$Q = 4\{[2(d^2 + 2d - 1 - c_r)I_T - (d^2 + (3-2c_r)d - 6c_r)I_L]^2 - 2(d+2)(d - c_r + 1)^2(I_T - I_L)^2\}. \tag{46}$$

From (38), (39), and (40), we can set

$$A\mathscr{S}_{ikkjmn} + B(\mathscr{S}_{ikjkmn} + \mathscr{S}_{ijkkmn})$$
$$= -\tfrac{1}{2}\{[d^2 + (c_r+1)d - 4]I_T + (c_r d + 4c_r + 2)I_L\}V_{ijmn} + \tfrac{1}{2}(dI_T + 2I_L)(\delta_{im}\delta_{jn} + \delta_{in}\delta_{jm})$$
$$+ (I_L - I_T)\delta_{ij}\delta_{mn} - \frac{1}{c_r}\{2(d - c_r + 1)I_T + [(c_r - 2)d + 4c_r - 2]I_L\}\delta_{ij}\hat{q}_m\hat{q}_n$$
$$= XV_{ijmn} + Y(\delta_{im}\delta_{jn} + \delta_{in}\delta_{jm}) + Z\delta_{ij}\delta_{mn} + W\delta_{ij}\hat{q}_m\hat{q}_n.$$

The second term of the integrand in (44) is given by

$$b_2 \int_q (\mathscr{X}^2)_{mm} = \frac{b_2 v^2}{d^2(d+2)^2} \overline{\theta}_i^a \theta_j^a \overline{\theta}_k^b \theta_l^b \int_q [XV_{ijmn} + Y(\delta_{im}\delta_{jn} + \delta_{in}\delta_{jm}) + Z\delta_{ij}\delta_{mn} + W\delta_{ij}\hat{q}_m\hat{q}_n]$$
$$[XV_{klmn} + Y(\delta_{km}\delta_{ln} + \delta_{kn}\delta_{lm}) + Z\delta_{kl}\delta_{mn} + W\delta_{kl}\hat{q}_m\hat{q}_n].$$

The integrand is

$$[XV_{ijmn} + Y(\delta_{im}\delta_{jn} + \delta_{in}\delta_{jm}) + Z\delta_{ij}\delta_{mn} + W\delta_{ij}\hat{q}_m\hat{q}_n][XV_{klmn} + Y(\delta_{km}\delta_{ln} + \delta_{kn}\delta_{lm}) + Z\delta_{kl}\delta_{mn} + W\delta_{kl}\hat{q}_m\hat{q}_n]$$
$$= 2X(X + 2Y)V_{ijkl} - 4c(1-c)X^2\hat{q}_i\hat{q}_j\hat{q}_k\hat{q}_l + 2[(1-c)X(Z+W) + YW](\hat{q}_i\hat{q}_j\delta_{kl} + \delta_{ij}\hat{q}_k\hat{q}_l)$$
$$+ 2Y^2(\delta_{ik}\delta_{jl} + \delta_{il}\delta_{jk}) + (4YZ + dZ^2 + 2ZW + W^2)\delta_{ij}\delta_{kl}.$$

Integrating this, we obtain

$$\int_q [XV_{ijmn} + Y(\delta_{im}\delta_{jn} + \delta_{in}\delta_{jm}) + Z\delta_{ij}\delta_{mn} + W\delta_{ij}\hat{q}_m\hat{q}_n][XV_{klmn} + Y(\delta_{km}\delta_{ln} + \delta_{kn}\delta_{lm}) + Z\delta_{kl}\delta_{mn} + W\delta_{kl}\hat{q}_m\hat{q}_n]$$
$$= \frac{1}{vd(d+2)c_r^2}[R(\delta_{ik}\delta_{jl} + \delta_{il}\delta_{jk}) + S\delta_{ij}\delta_{kl}],$$

where

$$R = d(d+2)c_r^2 \left[2Y^2 + \frac{1}{d}4X(X+2Y) - \frac{4c(1-c)X^2}{d(d+2)} - \frac{4(1+c)X(X+2Y)}{d(d+2)}\right] \tag{47}$$

$$S = d(d+2)c_r^2 \left\{\frac{4}{d}[(1-c)X(Z+W) + YW] + (4YZ + dZ^2 + 2ZW + W^2) - \frac{4(1+c)X(X+2Y) + 4c(1-c)X^2}{d(d+2)}\right\}. \tag{48}$$

From (45), (46), (47), and (48), we obtain

$$\overline{W_n[J]} = \int \mathscr{D}\theta \mathscr{D}\overline{\theta} e^{\tilde{C}\overline{\theta}_i^a \theta_i^a} \exp\left[\int_{k'} b_1(\mathscr{X}_{mm})^2 + b_2(\mathscr{X}^2)_{mm}\right]$$
$$= \int \mathscr{D}\theta \mathscr{D}\overline{\theta} e^{\tilde{C}\overline{\theta}_i^a \theta_i^a} \exp\left\{\frac{v}{d^3(d+2)^3 c_r^2}[(b_1 Q + b_2 S)\overline{\theta}_i^a \theta_i^a \overline{\theta}_j^b \theta_j^b + (b_1 P + b_2 R)(\overline{\theta}_i^a \theta_j^a \overline{\theta}_i^b \theta_j^b + \overline{\theta}_i^a \theta_j^a \overline{\theta}_j^b \theta_i^b)]\right\}$$
$$= \tilde{C}^{dn} \int \mathscr{D}\theta \mathscr{D}\overline{\theta} e^{\overline{\theta}_i^a \theta_i^a} \exp\left\{\frac{v}{d^3(d+2)^3 c_r^2}\left[\frac{A'}{\tilde{C}^2}\overline{\theta}_i^a \theta_i^a \overline{\theta}_j^b \theta_j^b + \frac{B'}{\tilde{C}^2}(\overline{\theta}_i^a \theta_j^a \overline{\theta}_i^b \theta_j^b + \overline{\theta}_i^a \theta_j^a \overline{\theta}_j^b \theta_i^b)\right]\right\},$$

where we changed the Grassmann variables as $\theta, \overline{\theta} \to \theta/\sqrt{\tilde{C}}, \overline{\theta}/\sqrt{\tilde{C}}$, and

$$
\begin{aligned}
A' = {} & -\frac{16}{(-1+c)^4}\{b_1(-1+c)^2[(-35+2c+c^2)I_L^2 - 2(-17+8c+c^2)I_L I_T + 8(-1+c)I_T^2] \\
& + b_2[(1+66c-38c^2+2c^3+c^4)I_L^2 - 2(5+14c-26c^2+6c^3+c^4)I_L I_T + 4(-1+c)^2(1+c)I_T^2]\} \\
& - \frac{8d}{(-1+c)^4}\{b_1(-1+c)^2[(-15-34c+5c^2)I_L^2 - 4(9-15c+4c^2)I_L I_T + (21-30c+13c^2)I_T^2] \\
& + b_2[(-13+66c+8c^2-34c^3+5c^4)I_L^2 - 4(2-27c+32c^2-19c^3+4c^4)I_L I_T \\
& + (-1-42c+84c^2-54c^3+13c^4)I_T^2]\} \\
& + \frac{4d^2}{(-1+c)^4}\{b_1(-1+c)^2[(-23+38c+c^2)I_L^2 - 4(-17+16c+c^2)I_L I_T + 8(-1+c)I_T^2] \\
& + b_2[(13-2c-58c^2+38c^3+c^4)I_L^2 - 4(-5+43c-50c^2+15c^3+c^4)I_L I_T + 4(1-5c+c^2+c^3)I_T^2]\} \\
& + \frac{d^3}{(-1+c)^3}\{8b_1(-1+c)^2[2(1+c)I_L^2 - 8cI_L I_T + 7(-1+c)I_T^2] \\
& + 4b_2[(-1-7c+4c^2+4c^3)I_L^2 - 8(1-2c-2c^2+2c^3)I_L I_T + (-5+41c-42c^2+14c^3)I_T^2]\} \\
& + \frac{4d^4}{(-1+c)^2}\{b_1(-1+c)^2(I_L-2I_T)^2 + b_2[c^2(I_L-2I_T)^2 + 4c(I_L-2I_T)I_T + 2I_T^2]\}
\end{aligned}
\tag{49}
$$

and

$$
\begin{aligned}
B' = {} & \frac{16}{(-1+c)^2}[(-5+c)I_L - 2(-1+c)I_T][b_1(-5+c)I_L - 2b_1(-1+c)I_T + b_2(-3+c)I_L - 2b_2(-1+c)I_T] \\
& + \frac{16d}{(-1+c)^4}\{b_1(-1+c)^2[-2(-5+c)I_L^2 + (11-4c+c^2)I_L I_T - 2(3-4c+c^2)I_T^2] \\
& + b_2[-2(-12+9c-6c^2+c^3)I_L^2 + (-13+8c+8c^2-4c^3+c^4)I_L I_T - 2(-3+c)(-1+c)^2 c I_T^2]\} \\
& + \frac{4d^2}{(-1+c)^4}\{b_1(-1+c)^2[4I_L^2 + 4(8-7c+c^2)I_L I_T + (1+10c-7c^2)I_T^2] \\
& + b_2[2(19-6c+3c^2)I_L^2 + 4(10-14c+19c^2-8c^3+c^4)I_L I_T + (-21+38c-36c^2+26c^3-7c^4)I_T^2]\} \\
& + \frac{8d^3}{(-1+c)^4}\{b_1(-1+c)^3 I_T[-2I_L + (-3+c)I_T] \\
& + b_2[2I_L^2 + (15-16c+7c^2-2c^3)I_L I_T + c(1+5c-5c^2+c^3)I_T^2]\} \\
& + \frac{2d^4}{(-1+c)^3} I_T[2b_1(-1+c)^3 I_T - 8b_2 I_L + b_2(-11+7c-6c^2+2c^3)I_T] \\
& + \frac{4d^5}{(-1+c)^2} b_2 I_T^2
\end{aligned}
\tag{50}
$$

Using these results, we can further simplify $\overline{W_n[J]}$ as follows:

$$
\begin{aligned}
\overline{W_n[J]} &= \tilde{C}^{dn} \int \mathscr{D}\theta \mathscr{D}\overline{\theta}\, e^{\overline{\theta}_i^a \theta_i^a} \exp[\tilde{A}\overline{\theta}_i^a \theta_i^a \overline{\theta}_j^b \theta_j^b + \tilde{B}(\overline{\theta}_i^a \theta_j^a \overline{\theta}_i^b \theta_j^b + \overline{\theta}_i^a \theta_j^a \overline{\theta}_j^b \theta_i^b)] \\
&= \tilde{C}^{dn} \sum_{m=0} \frac{\tilde{A}^m}{m!} \int \mathscr{D}\theta \mathscr{D}\overline{\theta} (\overline{\theta}_i^a \theta_i^a)^{2m} e^{\overline{\theta}_i^a \theta_i^a} \exp[\tilde{B}(\overline{\theta}_i^a \theta_j^a \overline{\theta}_i^b \theta_j^b + \overline{\theta}_i^a \theta_j^a \overline{\theta}_j^b \theta_i^b)] \\
&= \tilde{C}^{dn} \sum_{m=0} \frac{\tilde{A}^m}{m!} \int \mathscr{D}\theta \mathscr{D}\overline{\theta} \left(\frac{d}{dt}\right)^{2m} e^{t\overline{\theta}_i^a \theta_i^a}\bigg|_{t=1} \exp[\tilde{B}(\overline{\theta}_i^a \theta_j^a \overline{\theta}_i^b \theta_j^b + \overline{\theta}_i^a \theta_j^a \overline{\theta}_j^b \theta_i^b)] \\
&= \tilde{C}^{dn} e^{\tilde{A}\frac{d^2}{dt^2}}\bigg|_{t=1} \int \mathscr{D}\theta \mathscr{D}\overline{\theta}\, e^{t\overline{\theta}_i^a \theta_i^a} \exp[\tilde{B}(\overline{\theta}_i^a \theta_j^a \overline{\theta}_i^b \theta_j^b + \overline{\theta}_i^a \theta_j^a \overline{\theta}_j^b \theta_i^b)],
\end{aligned}
$$

where $\tilde{A} = vA'/d^3(d+2)^3 c_r^2 \tilde{C}^2$ and $\tilde{B} = vB'/d^3(d+2)^3 c_r^2 \tilde{C}^2$. By the Hubbard-Stratonovich transformation, we obtain

$$
\begin{aligned}
\int \mathscr{D}\theta \mathscr{D}\overline{\theta}\, e^{t\overline{\theta}_i^a \theta_i^a} \exp[\tilde{B}(\overline{\theta}_i^a \theta_j^a \overline{\theta}_i^b \theta_j^b + \overline{\theta}_i^a \theta_j^a \overline{\theta}_j^b \theta_i^b)] &= \int \mathscr{D}s\, e^{-s_{ij}s_{ij}} \int \mathscr{D}\theta \mathscr{D}\overline{\theta} \exp\left\{\left[t\delta_{ij} + \sqrt{2\tilde{B}}(s_{ij}+s_{ji})\right]\overline{\theta}_i^a \theta_j^a\right\} \\
&= \int \mathscr{D}s\, e^{-s_{ij}s_{ij}} \det{}^n\left[t\delta_{ij} + \sqrt{2\tilde{B}}(s_{ij}+s_{ji})\right].
\end{aligned}
$$

Taking the limit of $n \to 0$, we have

$$\lim_{n \to 0} \frac{\overline{W_n[J]}}{n} = e^{\tilde{A}\frac{d^2}{dt^2}}\Big|_{t=1} \int \mathscr{D}s\, e^{-s_{ij}s_{ij}} \lim_{n \to 0} \frac{1}{n} \tilde{C}^{dn} \left\{ \det^n \left[ t\delta_{ij} + \sqrt{2\tilde{B}}(s_{ij} + s_{ji}) \right] - 1 \right\}$$

$$= e^{\tilde{A}\frac{d^2}{dt^2}}\Big|_{t=1} \int \mathscr{D}s\, e^{-s_{ij}s_{ij}} \log \det \left[ t\tilde{C}\delta_{ij} + \sqrt{2\tilde{C}^2\tilde{B}}(s_{ij} + s_{ji}) \right]$$

$$= e^{\tilde{A}\frac{d^2}{dt^2}}\Big|_{t=1} \int \prod_{i \leq j} \left[ \frac{ds'_{ij}}{\sqrt{2\pi}^{2d}} \right] e^{-\mathrm{tr}\hat{S}'^2/2} \log \det \left( t\tilde{C}\hat{I} + 2\sqrt{\tilde{C}^2\tilde{B}}\hat{S}' \right),$$

where $\hat{S}'$ is a symmetric matrix. Diagonalizing this symmetric matrix, we obtain

$$\lim_{n \to 0} \frac{\overline{W_n[J]}}{n} = e^{\tilde{A}\frac{d^2}{dt^2}}\Big|_{t=1} \int \left( \prod_{i=1}^{d} dx_i \right) |V_d(\{x_i\})| e^{-\sum_{i=1}^{d} x_i^2/2} \sum_{i=1}^{d} \log \left( \tilde{C}t + 2\sqrt{\tilde{C}^2\tilde{B}}x_i \right),$$

where $V_d(\{x_i\})$ is the Vandermonde polynomial, and we ignored irrelevant coefficients. Using the eigenvalue distribution $\rho(x)$ of the Gaussian orthogonal ensemble (GOE), this integral simplifies to

$$\lim_{n \to 0} \frac{\overline{W_n[J]}}{n} = e^{\tilde{A}\frac{d^2}{dt^2}}\Big|_{t=1} \int dx\, \rho(x) \log \left( \tilde{C}t + 2\sqrt{\tilde{C}^2\tilde{B}}x \right)$$

$$= \int dx\, \rho(x) e^{\frac{\tilde{A}}{4\tilde{B}}\frac{d^2}{dx^2}} \log \left( \tilde{C} + 2\sqrt{\tilde{C}^2\tilde{B}}x \right)$$

$$= \int dx \left[ e^{\frac{\tilde{A}}{4\tilde{B}}\frac{d^2}{dx^2}} \rho(x) \right] \log \left( \tilde{C} + 2\sqrt{\tilde{C}^2\tilde{B}}x \right).$$

### 3. Small-$\mu$ limit

To simplify (49) and (50), we assume $\mu \ll 1$. When we assume $\mu^E = \mathscr{O}(\mu)$ and $\lambda^E = \mathscr{O}(1)$, we see $I_T = \mathscr{O}(\mu^{-1})$ and $I_L = \mathscr{O}(1)$, and hence we obtain $I'_T = \mu I_T = \mathscr{O}(1)$. In this limit, we have

$$A' = -\frac{2b_2 d^2 I'_T{}^2}{\mu^6} + \mathscr{O}(\mu^{-5}) \tag{51}$$

$$B' = \frac{b_2 d^3 I'_T{}^2}{\mu^6} + \mathscr{O}(\mu^{-5}) \tag{52}$$

$$\tilde{C} = 1 + v\left(1 - \frac{1}{d}\right)\left(1 - \frac{\mu^E}{\mu}\right) I'_T + \frac{v}{d}(1 - \lambda^E) I_L + \mathscr{O}(\mu). \tag{53}$$

Note that the relation $A' = -(2/d)B'$ always holds when we ignore higher order terms. Moreover, in this limit, $\lambda^E$ only enters in $\tilde{C}$. Thus, we can easily obtain $\lambda^E = 1 + \mathscr{O}(\mu)$. As a result, we have

$$\lim_{n \to 0} \frac{\overline{W_n[J]}}{n} \simeq \int dx \left[ e^{-\frac{1}{2d}\frac{d^2}{dx^2}} \rho(x) \right] \log \left[ 1 + \left(1 - \frac{1}{d}\right)\left(1 - \frac{\mu^E}{\mu}\right) v I'_T + \sqrt{ev}I'_T x \right]$$

$$= \int dx \left[ e^{-\frac{1}{2d}\frac{d^2}{dx^2}} \rho(x) \right] \log \left[ \frac{1}{\sqrt{ev}I'_T} + \frac{1}{\sqrt{e}}\left(1 - \frac{1}{d}\right)\left(1 - \frac{\mu^E}{\mu}\right) + x \right] + \log I'_T. \tag{54}$$

where

$$e = \frac{4b_2}{v(d+2)^3\mu^4} = \frac{4}{v(d+2)^3\mu^4 s_1}. \tag{55}$$

The distribution $\kappa(x) \equiv e^{-\frac{1}{2d}\frac{d^2}{dx^2}} \rho(x)$ can be rewritten as a convolution of $\rho(x)$ and a Gaussian

$$\kappa(x) = \int \frac{dl}{2\pi} e^{\frac{l^2}{2d}} e^{-ilx} \int dx' e^{ilx'} \rho(x')$$

$$= \int \frac{dl}{2\pi} \int dx' e^{-lx'} \rho(x') e^{-\frac{l^2}{2d}} e^{lx}$$

$$= \int dx' \rho(x') \frac{1}{\sqrt{2\pi/d}} \exp\left[ -\frac{(x - x')^2}{2/d} \right]. \tag{56}$$

Differentiating (54) with respect to $J_T$, the self-consistent equation for $\mu^E$ reads

$$vI'_T = \int dy \frac{\kappa(x)}{\frac{1}{vI'_T} + \left(1 - \frac{1}{d}\right)(1 - y) + \sqrt{e}x},$$

(57)

where $y = \mu^E/\mu$.

Before solving (57), we compare it with the EMT equation for spring networks. Further transformations of (57) yields

$$0 = \int dx \frac{\kappa(x)\left[\left(1 - \frac{1}{d}\right)(\mu^E - \mu) - \sqrt{e}\mu x\right]}{1 - vI_T\left[\left(1 - \frac{1}{d}\right)(\mu^E - \mu) - \sqrt{e}\mu x\right]}$$

and

$$vI_T = v \int_q \frac{q^2}{\mu^E q^2 - \rho\omega^2} = \frac{1}{\mu^E}\left(1 + \rho\omega^2 v \int_q \frac{1}{\mu^E q^2 - \rho\omega^2}\right).$$

When we denote the variance of $x$ by $\sigma_v^2$, a new random variable $\mu_r \equiv \mu + \sqrt{e}\mu x/(1 - 1/d)$ also follows the same distribution $\kappa(x)$ with the mean $\mu$ and the variance $e\mu^2\sigma_v^2/(1 - 1/d)^2$. Thus, we have

$$0 = \int d\mu_r \sigma(\mu_r) \frac{\mu^E - \mu_r}{1 - \frac{\mu^E - \mu_r}{\mu^E}\left(1 - \frac{1}{d}\right)\left(1 + \rho\omega^2 v \int_q \frac{1}{\mu^E q^2 - \rho\omega^2}\right)}.$$

(58)

This is the same equation as the one in [16]. To emphasize the correspondence, we change the notation as follows: $\mu^E \to k^{\parallel}$ and $\mu_r \to k_\alpha$, and set $\mathrm{tr}\mathscr{G}(0,\omega) = dv\int_q \frac{1}{\mu^E q^2 - \rho\omega^2}$ and $2d/z_0 = 1 - 1/d$. Using these notations, we have

$$0 = \int dx \sigma(k_\alpha) \frac{k^{\parallel} - k_\alpha}{1 - \frac{k^{\parallel} - k_\alpha}{k^{\parallel}}\frac{2d}{z_0}\left[1 + \frac{\rho\omega^2}{d}\mathrm{tr}\mathscr{G}(0,\omega)\right]}.$$

(59)

This is exactly the self-consistent equation of the standard EMT for a disordered spring. Thus, the random variable $x$ can be interpreted as a normalized space-dependent shear modulus. This justifies previous studies of the EMT applied to simple spring networks and determine the natural distribution $\kappa(x)$ that the shear modulus of glasses follows, i.e., the convolution of the Gaussian distribution and the GOE eigenvalue distribution, at least without stability conditions.