# Peer review of "Random quench predicts universal properties of amorphous solids"

_SciPost Physics_

## Round 2 · Referee Report · Anonymous (Referee 1) · 2021-10-15

Report
In this work, the authors examine a minimal model of an overdamped quench of an amorphous solid into a so-called inherent state (IS), enforcing mechanical equilibrium but not stability per se. They show that even when starting from a initially ‘featureless’ distribution of stress, it is possible to reproduce both the random stress and the fluctuations of elastic moduli associated to an IS, and from their combination to recover the universal vibrational anomalies observed in glasses.
More specifically: 1) They first assume that the system has initially a random ‘quench stress’, sorted out from a Gaussian distribution of the stress tensor field with a zero correlation length. 2) Then they rely on previous work by Di Donna and Lubensky (Ref. [35] in the paper) who provided the expressions for the stress in the new IS and the elastic modulus tensor around the IS after such an overdamped quench (at least at leading order in the displacement field along the quench). Admittedly these expressions were obtained by assuming that the quench stress (i.e. the initial condition) had a zero mean, but the authors here argued that they can be easily adapted to the more general case with non-zero mean by a simple translation of the Lamé moduli which parametrize the elasticity. 3) From there, they were able to compute explicitly the distribution of elastic modulii and of stress, as well as their spatial correlations. In two and three dimensions, they used in particular that the stress tensor could be conveniently represented in terms of a gauge field, which was previously shown to relate to long-range correlations in the stress field. 4) They were able to compute the density of vibrational modes within the effective medium theory (EMT), introduced as a small-disorder mean-field approximation. For that, in the infinite-dimensional limit they relied on random matrix theory results, specifically on the spectrum of the Gaussian Orthogonal Ensemble (GOE) matrix. They further discussed the corrections expected in finite dimension, and their implications for systems in the thermodynamic limit versus with finite size.
These results emphasize that it is not sufficient to assume some random stress distribution around an IS to reproduce the anomalous vibrational spectrum of glasses, whereas all relevant features (random stress, fluctuations of elastic modulii and the expected vibrational spectrum) can emerge on their own already following an overdamped quench. I think that this is an important contribution to the field, and as such I recommend the publication of the manuscript in SciPost.
Requested changes
I listed thereafter some comments or typos that I would like to invite the authors to consider. As a general remark, the manuscript was quite dense to read, and I do not know how accessible it could be for non-experts, though a few additions could change that, I believe. - In the abstract, « a universal field-theoretic model of an overdamped quench »: although a model can reproduce universal properties, I think that stating that the model is universal is a bit of an overstatement, I would suggest to remove the « universal » there. - The notion of « quench » is assumed to be known beforehand, and not really explained. To make this accessible to non-expert, a short explanation could be welcome. - In Eq. (1), shouldn’t the displacement field u_i(t) be a vector, by consistency with the notation used later in the text? - When introducing the Green function, it would be useful to have a short explanation and/or a reference as to why it is the right quantity to consider for accessing the vibrational spectrum. - In Eq. (2), what is the definition of the average with the brackets? I think it is missing. - At the end of page 2, it is assumed that we start from « an initial homogeneous elastic continuum with elastic constants $\tilde\lambda$ and $\tilde\mu$ »: it would be welcome to recall the explicit expression of the corresponding elastic modulus tensor, for non-expert readers. - Before Eq.(11), when the stress is expressed in terms of a gauge field, it might be useful to recall that the \epsilon correspond to the Levi-Civita symbol (if it is indeed so, I initially thought it was the strain...). - In Eq.(11), could you comment a bit this distribution? What about its normalization? How does the most probable configuration look like in real space for instance? - After Eq.(15), it is written that « We emphasize that \sigma is a function of […] and thus these distributions are nontrivial ». I do not really understand this statement here, since the expressions provided have no explicit dependence on the gauge field left. So it is as if we could forget about the gauge field once we finished the computation. Could you comment a bit more? - Before Eq.(16), the shear modulus fluctuation has a denominator (d^3+d^2-2d); is there any physical meaning/intuition to this specific combination of the dimension d? - As a pure cosmetic remark, Eq.(17) could be formatted in a clearer way, aligning the expressions on the equality signs at least. - In Eq. (19), the Green function is decomposed into G_T and G_L: what do T and L stand for? - Before Eq.(20), it is stated that the results are reported for the limit $\mu \ll 1$: could you comment on the physical meaning of this limit? - In Eq.(21), if I understood correctly, \sigma(x) is not stress, or is it? If this is not the case, this notation is a bit confusing. If it is a stress, however, a comment on that would be welcome. - After Eq.(21), it is emphasized that « the GOE matrix whose spectrum appears in (21) is not put into the model, but emerges from its solution ». Is it possible to track this fact down to some specific assumptions that were made along the way to reach this result? Can we say something about how we could depart from this by tuning some specific ingredient? - Just before the conclusion, and in the conclusion itself, it is emphasized that local stability is not enforced. Could you explicit a bit the difference between this local stability and (local?) mechanical equilibrium? This would clearly be useful for non-experts. - In Appendix 2.1, just after « The probability distribution function of $\delta K$ can be written as […] », shouldn’t $\tau$ be denoted $\hat{\tau}$ if it is a tensor (as it was in Appendix 1)? - After Eq.(25), just after « As in the case of $\delta K$, the probability distribution […] », there is a $\hat{\bar{\sigma}}$: if it is the tensor of the quench stress, shouldn’t it be instead $\hat{\tilde{\sigma}}$? - The very last paragraph of the Appendix, the sentence « the random variable $x$ can be interpreted as a normalized space-dependent shear modulus » is a comment that I would have found very useful to have in the main text.
In this work, the authors examine a minimal model of an overdamped quench of an amorphous solid into a so-called inherent state (IS), enforcing mechanical equilibrium but not stability per se. They show that even when starting from a initially ‘featureless’ distribution of stress, it is possible to reproduce both the random stress and the fluctuations of elastic moduli associated to an IS, and from their combination to recover the universal vibrational anomalies observed in glasses.
More specifically: 1) They first assume that the system has initially a random ‘quench stress’, sorted out from a Gaussian distribution of the stress tensor field with a zero correlation length. 2) Then they rely on previous work by Di Donna and Lubensky (Ref. [35] in the paper) who provided the expressions for the stress in the new IS and the elastic modulus tensor around the IS after such an overdamped quench (at least at leading order in the displacement field along the quench). Admittedly these expressions were obtained by assuming that the quench stress (i.e. the initial condition) had a zero mean, but the authors here argued that they can be easily adapted to the more general case with non-zero mean by a simple translation of the Lamé moduli which parametrize the elasticity. 3) From there, they were able to compute explicitly the distribution of elastic modulii and of stress, as well as their spatial correlations. In two and three dimensions, they used in particular that the stress tensor could be conveniently represented in terms of a gauge field, which was previously shown to relate to long-range correlations in the stress field. 4) They were able to compute the density of vibrational modes within the effective medium theory (EMT), introduced as a small-disorder mean-field approximation. For that, in the infinite-dimensional limit they relied on random matrix theory results, specifically on the spectrum of the Gaussian Orthogonal Ensemble (GOE) matrix. They further discussed the corrections expected in finite dimension, and their implications for systems in the thermodynamic limit versus with finite size.
These results emphasize that it is not sufficient to assume some random stress distribution around an IS to reproduce the anomalous vibrational spectrum of glasses, whereas all relevant features (random stress, fluctuations of elastic modulii and the expected vibrational spectrum) can emerge on their own already following an overdamped quench. I think that this is an important contribution to the field, and as such I recommend the publication of the manuscript in SciPost.
We thank the referee for reading our manuscript carefully and appreciating the results.
I listed thereafter some comments or typos that I would like to invite the authors to consider. As a general remark, the manuscript was quite dense to read, and I do not know how accessible it could be for non-experts, though a few additions could change that, I believe.
We thank the referee for their useful comments. We believe that the manuscript has improved considerably.
- In the abstract, « a universal field-theoretic model of an overdamped quench »: although a model can reproduce universal properties, I think that stating that the model is universal is a bit of an overstatement, I would suggest to remove the « universal » there.
- The notion of « quench » is assumed to be known beforehand, and not really explained. To make this accessible to non-expert, a short explanation could be welcome.
- In Eq. (1), shouldn’t the displacement field u_i(t) be a vector, by consistency with the notation used later in the text?
- When introducing the Green function, it would be useful to have a short explanation and/or a reference as to why it is the right quantity to consider for accessing the vibrational spectrum.
Thanks. We agree with the comments and revised the manuscript based on them.
- In Eq. (2), what is the definition of the average with the brackets? I think it is missing.
This bracket denoted the thermal average, but since it is not essential in the manuscript, we removed it (also taking into account a comment from Referee 2).
- At the end of page 2, it is assumed that we start from « an initial homogeneous elastic continuum with elastic constants $\tilde \lambda$ and $\tilde \mu$ » : it would be welcome to recall the explicit expression of the corresponding elastic modulus tensor, for non-expert readers.
- Before Eq.(11), when the stress is expressed in terms of a gauge field, it might be useful to recall that the \epsilon correspond to the Levi-Civita symbol (if it is indeed so, I initially thought it was the strain...).
We thank the referee for these comments. We revised the manuscript.
- In Eq.(11), could you comment a bit this distribution? What about its normalization? How does the most probable configuration look like in real space for instance?
This distribution was thoroughly investigated in Ref. [37,38]. Thus we refrained from repeating the same analysis and just wrote that “They predict anisotropic long-range correlations in the stress field, as discussed at length in [37, 38].”
- After Eq.(15), it is written that « We emphasize that \sigma is a function of […] and thus these distributions are nontrivial ». I do not really understand this statement here, since the expressions provided have no explicit dependence on the gauge field left. So it is as if we could forget about the gauge field once we finished the computation. Could you comment a bit more?
The point is that one cannot consider arbitrary functions $\sigma_{ij}(\vec{r})$ because they will not satisfy the constraints of mechanical equilibrium. The gauge fields are necessary to impose the constraints on the stress tensor. We added a sentence to explain it below Eq 15. The difference between an arbitrary stress distribution and one in mechanical equilibrium can be visualized in Fig 1 (there for the xy component).
- Before Eq.(16), the shear modulus fluctuation has a denominator (d^3+d^2-2d); is there any physical meaning/intuition to this specific combination of the dimension d?
In $d = 0$ and $d = 1$ there is no shear modulus, so the expression should become degenerate and the denominator should vanish in those dimensions (and it does). For larger $d$, it should have a definite sign, and it does. Beyond that, we don’t see any specific meaning to it.
- As a pure cosmetic remark, Eq.(17) could be formatted in a clearer way, aligning the expressions on the equality signs at least.
We thank the referee for this suggestion. We revised the manuscript.
- In Eq. (19), the Green function is decomposed into G_T and G_L: what do T and L stand for?
T and L stand for transverse and longitudinal, respectively. We added this sentence below Eq 19.
- Before Eq.(20), it is stated that the results are reported for the limit $\mu\ll1$: could you comment on the physical meaning of this limit?
We added a comment below Eq 20.
- In Eq.(21), if I understood correctly, \sigma(x) is not stress, or is it? If this is not the case, this notation is a bit confusing. If it is a stress, however, a comment on that would be welcome.
Thanks, indeed this is not stress. We changed the notation from $\sigma$ to $\kappa$.
- After Eq.(21), it is emphasized that « the GOE matrix whose spectrum appears in (21) is not put into the model, but emerges from its solution ». Is it possible to track this fact down to some specific assumptions that were made along the way to reach this result? Can we say something about how we could depart from this by tuning some specific ingredient?
This is an interesting but difficult question. We guess that the GOE matrix emerges from the Gaussian distribution of the initial stress and the effective medium approximation; namely, the structure of the functional$ W_n[J]$ in Eq. (35).
Just before the conclusion, and in the conclusion itself, it is emphasized that local stability is not enforced. Could you explicit a bit the difference between this local stability and (local?) mechanical equilibrium? This would clearly be useful for non-experts.
Thanks, we elaborated on this. Briefly, a pencil standing on its head is in mechanical equilibrium, but it is unstable. Our model allows regions of local instability, which under real dynamics would likely relax to a stable configuration.
- In Appendix 2.1, just after « The probability distribution function of $\delta K$ can be written as […] », shouldn’t $\tau$ be denoted $\hat{\tau}$ if it is a tensor (as it was in Appendix 1)?
We used the same symbol $\tau$ in Appendices 1 and 2 because it is an auxiliary field for an integral representation of the delta function, but in Appendix 2.1, it is a scalar field because $\delta K$ is a scalar.
- After Eq.(25), just after « As in the case of $\delta K$, the probability distribution […] », there is a $\hat{\bar \sigma}$: if it is the tensor of the quench stress, shouldn’t it be instead $\hat{\tilde \sigma}$?
- The very last paragraph of the Appendix, the sentence « the random variable x can be interpreted as a normalized space-dependent shear modulus » is a comment that I would have found very useful to have in the main text.
We thank the referee for these comments. We revised the manuscript.
The authors have addressed all my comments, so I fully recommend publication.

Author: Masanari Shimada on 2021-12-03 [id 2005]
(in reply to Report 2 on 2021-10-25)We thank the referee for reading our manuscript carefully and appreciating the results. In the following we answered all the comments.
Yes, this interpretation is correct. We revised the manuscript, see the explanation after Eq. (2).
We thank the referee for this insightful comment. As the referee pointed out, we need to include fluctuations in elastic constants to construct a realistic model. However, it is almost impossible for the moment to treat them in addition to the stress distribution. The present model should be interpreted as a minimal model, which is not very realistic, but sufficient to reproduce important elastic properties of glasses. We mentioned the importance of the fluctuations of elastic constants in the conclusion of the revised manuscript.
We removed the word “overdamped” from the abstract.
In principle a field-theoretical model should hold on scales that are much larger than the relevant microscopic scale (here the particle size) and much smaller than the system-size scale.
We removed the bracket from Eq (2).
Yes. The steepest descent algorithm can be implemented simply by the Langevin equation with the noise term dropped, which is exactly the overdamped dynamics considered by Di Donna and Lubensky in Ref. [35].
It is difficult to answer this question clearly, but the Gaussian distribution of the initial stress is one of the main assumptions in our model and thus we guess that most of the results depend on it. Conversely, since our results are consistent with numerical simulations or other theories, we think that the assumption of the Gaussian distribution is justified.
We expanded the discussion in the Appendix to better explain our method. In particular it mimics what was previously considered for lattice systems. We thus cite a review paper by Kirkpatrick in Ref. [54].
Yes, we added an explanation to the revised manuscript.
To our knowledge, there seems to be no correct study on the exactness of the effective medium approximation in elasticity models. ( Ref [40] attempts to derive the EMT/CPT for elasticity, but unfortunately it contains mistakes. In particular the authors use the replica method (which is fine) but going from Eq 18 to Eq 22 they replicate the disorder, which is incorrect.) We modified the explanation of DMFT on page 4.
In the case of crystals, for example, a few tens of particles are sufficient for the Debye model to be applicable as discussed in PNAS 106, 16907 (2009). It would be natural to expect similar behavior for glasses.
If the “defect term” $\Delta A$ is small, this yields a systematic perturbative expansion. In the manuscript, after Eq. (28), we performed a resummation to obtain the T-matrix expansion, see the revised manuscript.

---

## Round 2 · Referee Report · Anonymous (Referee 2) · 2021-10-25

Report
The paper is concerned with a subject of current interest, makes an important contribution to this subject and it is well written (especially concerning rather difficult techniques used). I recommend that it is accepted after the authors consider the following comments and make the appropriate changes in their manuscript.
The model described in the paper can be treated as a thought experiment and if this were the authors' only goal, they definitely succeeded. I have an impression, however, that they would like to connect their model to physically realistic simulations of particle-based systems. I strongly recommend that this connection is made more explicit and that the limitations of the present thought quench procedure are discussed.
It seems that the authors describe an instantaneous quench of a high temperature liquid to zero temperature (i.e. setting all the velocities to zero). This quench results in a structure that is not in a mechanical equilibrium, with a distribution of the stress (quench stress) that is inherited from the liquid. This state then relaxes to a mechanical equilibrium with overdamped equations of motion involving short time elastic constants of the liquid. The resulting equilibrium state (IS) is characterized by a distribution of elastic moduli and its own distribution of the stress. If this brief interpretation of the authors' thought experiment is correct, it should be made more explicit in the paper.
If this interpretation is correct, however, it seems that the authors omitted one feature of the liquid. If they want to treat the liquid as disordered (albeit not quenched) medium, which is suggested by using the liquid's local stress distribution as a "generator" of the quench stress, they should also include the fact that short-time elastic constants can be defined locally and that the distribution of these local elastic constants will be inherited by the quenched structure. Thus, there could/should? be two different sources of disorder: random stresses and random elastic constants of the initial liquid.
Parenthetically, I should mention that computer simulations of Barrat and collaborators did show that in zero temperature glasses both Born elastic constants (which seem to correspond to the initial elastic constants) and the non-affine contributions to elastic constants can be defined locally and that the fluctuations of these two contributions are comparable.
Requested changes
In addition to the general comment made in the report I suggest that the authors consider the following remarks: 1) I would remove "overdamped" from the abstract; "overdamped quench" is not a commonly used phrase and it may confuse some readers. 2) Could the authors define the spatial scale on which their model is supposed to be valid? I.e., quantitatively, what is "mesoscale"? 3) If brackets in Eq. (2) denote equilibrium average, the resulting average stress in a liquid is spatially uniform and diagonal. In other words, this equation is confusing. 4) Using the authors' language, overdamped quench seems to be the same as steepest descend? 5) I was wondering how much do the conclusions of the paper depend on the Gaussian character of the quench stress distribution, Eq. (3)? 6) Could the authors provide a reference to the version of EMT they are using? Preferably a reference different from their own past papers. 7) If the large wavevector cutoff is of the order of inverse particle size, the correlation volume on p. 5 is of the order of the particle size? Could the authors clarify? 8) The reference supporting the exact character of EMT in the d->oo limit is concerned with DMFT for correlated electrons, which differs from the present theory in many respects. Could the authors provide an alternative reference? 9) Can a long-wavelength theory be expected to describe the DOS of systems of a few hundred particles? 10) Could the authors clarify the physical meaning of the expansion between Eqs. (27) and (28)?

---

## Editorial Decision

unknown